# HIV-1 diverts cortical actin for particle assembly and release

Rayane Dibsy[1], Erwan Bremaud [1], Johnson Mak [2], Cyril Favard [1] & Delphine Muriaux [1]✉

Enveloped viruses assemble and bud from the host cell membranes. Any role of cortical actin in these processes have often been a source of debate. Here, we assessed if cortical actin was involved in HIV-1 assembly in infected CD4 T lymphocytes. Our results show that preventing actin branching not only increases HIV-1 particle release but also the number of individual HIV-1 Gag assembly clusters at the T cell plasma membrane. Indeed, in infected T lymphocytes and in in vitro quantitative model systems, we show that HIV-1 Gag protein prefers areas deficient in F-actin for assembling. Finally, we found that the host factor Arpin, an inhibitor of Arp2/3 branched actin, is recruited at the membrane of infected T cells and it can associate with the viral Gag protein. Altogether, our data show that, for virus assembly and particle release, HIV-1 prefers low density of cortical actin and may favor local actin debranching by subverting Arpin.

Cortical actin is present below the plasma membrane and is involved in various major cell functions such as cell morphology[1], cell division[2], T cell activation[3] and cell tension[4]. Cortical actin is highly regulated by many cell factors, notably the major RhoGTPases, which can modulate cortical actin shape and function via a variety of signaling cascades and interacting partners (reviewed in[5–7]). Interestingly, some actin cofactors have been reported to play a role in the assembly and release of several viruses[8–10], including the human immunodeficiency virus type 1 (HIV-1). HIV-1 assembles and buds from the plasma membrane of host CD4⁺ T lymphocytes, the major targeted cells by the virus. The assembly of HIV-1 particles occurs at the inner surface of the plasma membrane of T lymphocytes and is mediated by the viral Gag polyprotein, which contains three major domains involved in this process. The matrix domain 'MA' is responsible of membrane binding via phosphatidylinositol 4,5-biphosphate (PI(4,5)P₂) interactions[11,12], the capsid domain 'CA' allows for Gag-Gag direct interactions, and the nucleocapsid domain, 'NC', binds the viral genomic RNA. We have recently shown that, during assembly, HIV-1 tuned the lipid composition of the membrane by generating nanodomains enriched in PI(4,5)P2 and Cholesterol, in order to facilitate the formation of new particles[13,14].

Underneath this membrane, distinct cortical actin nanostructures have been visualized at the viral buds using cryo-electron microscopy[15], reporting the presence of bundles of filaments or branched actin. Additionally, several actin cofactors have been shown to be involved in viral particle production[16,17]. However, the role of cortical actin in the late phases of HIV-1 replication cycle, i.e., particle assembly and release, is still controversial. As an example, treatment with drugs targeting the actin polymerization showed contradictory results[18–20] that could be explained by the use of different cell lines and different experimental conditions (reviewed in[21]).

The Arp2/3 complex is the mediator of branched actin, an important part of cortical actin. The Arp2/3 complex has been shown to regulate actin nucleation at the cell-cell contact surface during the transfer of HIV-1 particles from human dendritic cells to T lymphocytes[22]. Arp2/3 nucleation is upregulated by the activation of the two main RhoGTPases, CDC42 and Rac1. The activation of this nucleation complex is a standard process guaranteed by nucleation-promoting factors (NPFs), while its inhibition can be achieved in different ways, either by inhibiting the interaction between Arp2/3 and actin, or by blocking the direct interaction of NPFs activators, or by

---

[1]Institute of Research in Infectious disease of Montpellier (IRIM), University of Montpellier, UMR9004 CNRS Montpellier, France. [2]Institute for Glycomics, Griffith University, Brisbane, Australia. ✉e-mail: delphine.muriaux@irim.cnrs.fr

sequestering the complex as it is done by Coronin proteins[23], GMF[24,25], Gadkin or Arpin[26].

We previously showed, using siRNA mediated knock down, that the cortical actin signaling pathway IRSp53/Wave2/Arp2/3, regulated by Rac1, was involved in HIV-1 particle production in CD4[+] T lymphocytes[16] and that Rac1 was required for Gag plasma membrane attachment. HIV-1 Gag multimers locally generate a curvature when assembling at the cell plasma membrane leading to the formation and release of new viral particles. Indeed, we recently reported that the membrane curvature and actin signaling adaptor protein IRSp53 assists the completion of the viral bud generated by Gag[16,27].

Here, we further investigate the role of F-actin in HIV-1 assembly and reveal that HIV-1 assembly is fostered in membrane areas less dense in F-actin, enhanced upon F-actin debranching at the host T cell membrane and that, to favor HIV-1 particle formation, the viral Gag protein can hijack the host cell factor Arpin that inhibits Arp2/3 and prevents actin branching.

## Results

### Pharmacological drugs interfering with F-actin affect HIV-1 release in infected CD4[+] T lymphocytes

In order to assess any role of actin filament during HIV-1 particle production, we evaluated the effect of actin-related drugs on HIV-1 release from infected CD4[+] T lymphocytes. We first targeted actin filament regulation with known drugs that inhibit the main small RhoGTPases Rac1 and CDC42, ie. EHT1864 ([28,29]) and CDC42 inhibitor III[30], respectively, and that target directly actin depolymerization or polymerization, ie. latrunculin B[31] or jasplakinolide[32], respectively, in infected Jurkat T cells (Fig. 1) and primary blood T cells (Supplementary Fig. 2). To monitor and visualize virus assembly and release, ie. only the late steps of HIV-1 infection cycle, avoiding re-infection and entry, we used a single round virus, VSV-G pseudotyped HIV-1(i)GFP depleted from its Env glycoproteins (pVSV-G + pCMV-NL4.3(i)GFPΔEnv) for T cell infection (described in Supplementary Fig. 1). This internally GFP tagged virus is a powerful tool for investigating virus assembly mechanism, especially, using microscopy techniques[33]. To validate our VSV-G pseudotyped HIV-1(i)GFPΔEnv virus production and infectivity, we compared it to wild-type (WT) HIV-1. First, we compared viral particle release and showed that HIV-1 Gag as well as HIV-1 Gag(i)GFP release from productive model cell line was similar using Western Blot (Supplementary Fig. 1A, B). Secondly, we observed that Gag maturation was similar in both viruses by quantifying mature p24-Capsid in purified virions (Supplementary Fig. 1C). In addition, virus maturation of VSV-G pseudotyped HIV-1(i)GFPΔEnv virus was monitored by the release of internal GFP in virions (Supplementary Fig. 1D) (as described previously in[33]). Titrations of both viruses (in ng/ml of p24) were also equivalent between HIV-1 WT and HIV-1(i)GFP over 3 independent preparations (Supplementary Fig. 1E). Finally, VSV-G pseudotyped HIV-1(i)GFP was shown to efficiently infect Jurkat T cells (Supplementary Fig. 1F, H) for further following investigations throughout this study. These results confirmed the validity of our single round Gag(i)GFP virus to study HIV-1 assembly and release.

We then investigated any role of F-actin (de)polymerization by measuring the effect of latrunculin B and jasplakinolide on infected Jurkat T cells using the single round virus (Fig. 1). Interestingly, our results showed no significant decrease in viral release at low concentration (0.25 μM) of Latrunculin B together with a significant increase in viral release at higher concentration (1 μM) (Fig. 1A), unlike jasplakinolide treatment that showed a decrease in viral release with IC50 equal to 0.25 μM of jasplakinolide (Fig. 1A), as in infected peripheral blood lymphocytes (PBLs) (Supplementary Fig. 1B). Drug toxicity was evaluated by treating Jurkat T cells, for 24 hours, with a range of latrunculin B (0–100 μM) and jasplakinolide (0-20 μM). CC50 measured by cell viability assay was observed equal to 100 μM of latrunculin B and 2 μM of jasplakinolide respectively, confirming that

our experimental conditions were non-toxic (Fig. 1A). Since latrunculin B and jasplakinolide are known to depolymerize or to stabilize F-actin, respectively, we confirmed their effect in our cell system. For that, we quantified total F-actin intensity in Jurkat T cell treated with low and high concentrations of latrunclin B using Z projection imaging (Fig. 1B). Results showed F-actin depolymerization upon 24 h latrunculin B cell treatment. At a high concentration of latrunculin B (1 μM), we observed a strong decrease of cellular F-actin intensity ((0.07 ± 0.02). 10[4] a.u.; $n = 13$ different cells) as compared to the control ((1.92 ± 0.13). 10[4] a.u.; $n = 20$ different cells) (Fig. 1B graph). Cellular F-actin intensity with latrunculin B was also measured using flow cytometry in infected Jurkat T cells and quantification showed a decrease in cells labeled with phalloidin (with 5% in latrunculin B treated cells compared to 21% for the control) (Supplementary Fig. 3A), confirming F-actin depolymerization (Fig. 1B). In addition, we tested cellular F-actin stabilization upon jasplakinolide treatment using F-actin antibody staining, since jasplakinolide competed with phalloidin[34]. Flow cytometry analysis showed a 17% increase in F-actin mean intensity in cells treated with jasplakinolide compared to control (Supplementary Fig. 2A), confirming the effect of jasplakinolide in stabilizing F-actin in infected T cells. We also compared the level of infectivity in drug-treated and untreated Jurkat T cells using GFP fluorescence quantification by flow cytometry. As in the control (DMSO), 16% to 17% of the cell population positive for GFP was found in infected T cells treated with latrunculin B or jasplakinolide (Supplementary Fig. 3B), confirming that the changes in viral release were not due to a change in cellular Gag(i)GFP expression.

We finally confirmed any role of F-actin in HIV-1 infected activated PBLs that were purified from several healthy blood donors (Supplementary Fig. 2). PBLs were treated 24 hours post-infection with a range of drug concentrations, and viral release was measured 48 hours post-infection by monitoring p24 release (using p24-alpha-LISA immunoassay). As reported, we first checked that actin regulators targeting IRSp53, like Rac1 and CDC42, and subsequent F-actin regulation are, as expected, involved in HIV-1 particle release[16,27,35]. Indeed, results showed that inhibition of CDC42 with CDC42 inhibitor III (for a range of 0–400 μM) or Rac1 with EHT1864 (for a range of 0–200 μM), with cell viability superior to 70%, decreased HIV-1 release in a dose-dependent manner, for wild type HIV-1 infected PBLs with an IC50 equal to 150 μM of CDC42 inhibitor III and 50 μM of EHT1864. For VSV-G pseudotyped HIV-1(i)GFPdeltaEnv, the IC50 was equal to 200 μM of CDC42 inhibitor III and 50 μM of Rac1 inhibitor EHT1864 (Supplementary Fig. 2A).

In the context of PBLs, drugs that directly target actin filament (de)polymerization, latrunculin B and jasplakinolide, were further tested (Supplementary Fig. 2B). Both drugs decreased HIV-1 particle release from both wild-type and single-round HIV-1 infected PBLs, but with different efficacy. Latrunculin B treatment has a greater effect at low concentrations (0.25 μM) causing a maximum of half decrease in virus release, with a lesser effect at higher concentrations (1 μM) (Supplementary Fig. 2B). On the other hand, jasplakinolide treatment caused a more drastic decrease in virus release (76 ± 3%, $N = 5$ for WT infection, and 75 ± 1%, $N = 5$ for single round virus infection) at low concentration (0.25 μM), and this effect is sustained at higher concentrations unlike for latrunculin B (Supplementary Fig. 2B). These results indicated that HIV-1 infected PBLs were very sensitive to jasplakinolide treatment suggesting that stabilizing F-actin is deleterious to viral production. However, the effect of F-actin depolymerization, using latrunculin B treatment, on HIV-1 release from infected PBLs could vary depending on drug concentrations when quantified using p24 alphaLISA immunoassay.

Overall, our results on VSV-G pseudotyped HIV-1(i)GFPΔEnv infected Jurkat T cells and PBLs, indicated that stabilization of F-actin by jasplakinolide decreased viral release while F-actin depolymerization by latrunculin B showed different effect based on the

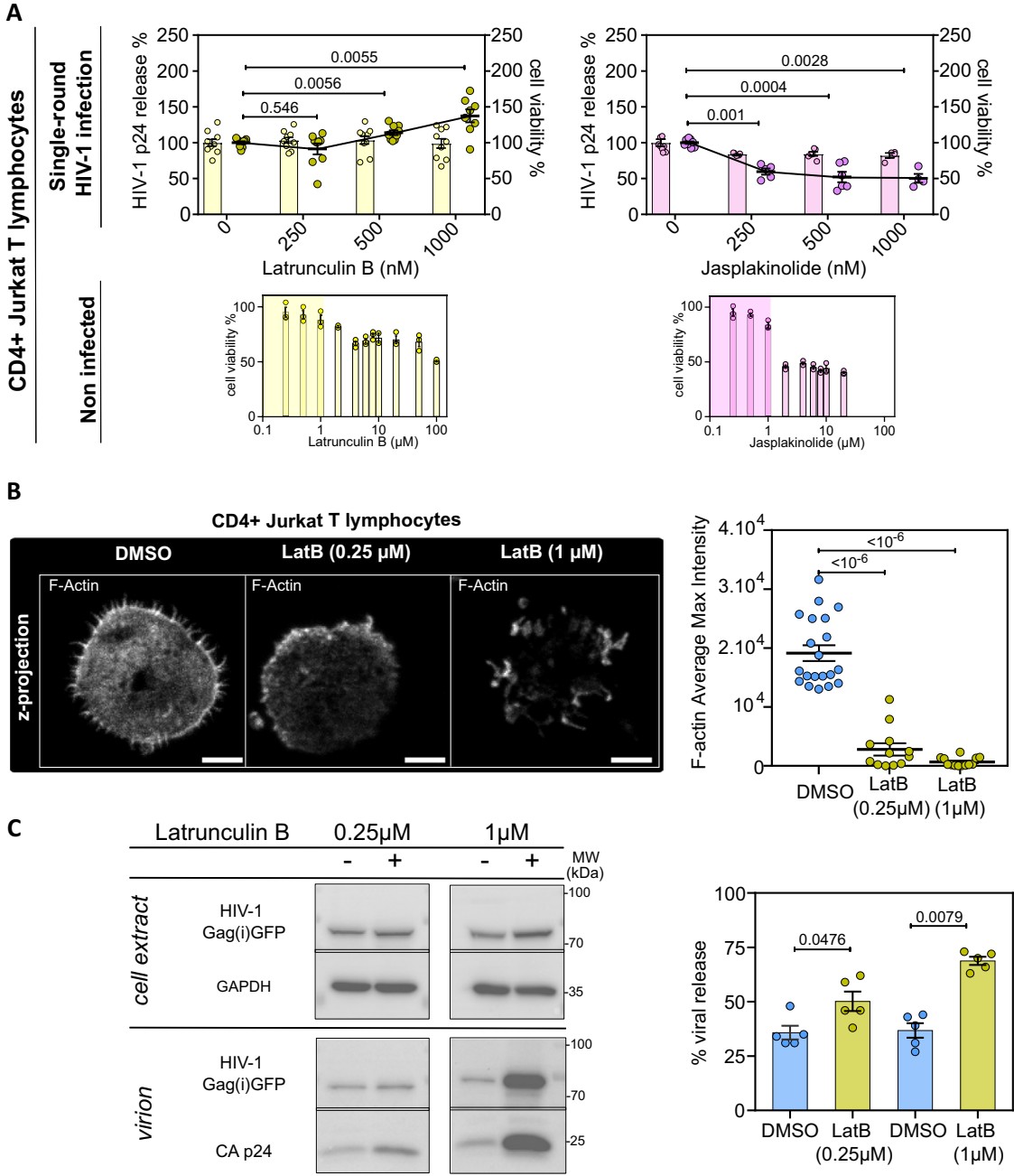

**Fig. 1 | Drugs targeting the (de)polymerization of F-actin alter HIV-1 release in infected CD4 + T lymphocytes.** A Upper panel: Left Y axis represents the relative percentage of HIV-1 p24 release (scatter dots, means are linked as a curve) from CD4+ Jurkat T lymphocytes infected with VSV-G pseudotyped HIV-1(i)GFPΔEnv single round virus and treated 24 hours post-infection with 0, 0.25, 0.5, and 1 μM of latrunculin B (in yellow) and 0, 0,25, 0,5 and 1 μM of jasplakinolide (in purple). The right Y axis represents the percentage of cell viability (bars) normalized to the control (zero drug) measured with 96AQ CellTiter. (*N* = 3 independent experiments; 4≤n ≤ 9). Lower panel: Graphs represent the percentage (%) of cell viability of non-infected CD4+ Jurkat T lymphocytes treated for 24 hours with 0, 0.25, 0.5, 2, 4, 6, 8, 10, 20, 50, and 100 μM of latrunculin B (LatB) and 0, 0.25, 0.5, 2, 4, 6, 8, 10, and 20 μM of jasplakinolide (*n* = 3 technical replicates). Yellow and pink zones limit respectively the range of latrunculin and jasplakinolide concentrations having no or

low toxicity. **B** Z projection of CD4+ Jurkat T lymphocytes treated with DMSO, 0.25 μM, and 1 μM of latrunculin B, stained with phalloidin Alexa Fluor 647 representing F-actin in gray. The scale bar is equal to 5 μm. Dot plot representing the quantification of F-actin intensity in each condition corresponding to the Z projection images. (DMSO: *n* = 20 cells; 0.25 μM latrunculin B: *n* = 12 cells; 1 μM latrunculin B: *n* = 13 cells). **C** Immunoblot of cell extract showing HIV-1(i)GFP Gag band ( - 82 kDa). GAPDH (36 kDa) is used as a loading control from the same blot. Purified virus release shows HIV-1(i)GFP Gag, as well as cleaved HIV-1 CAp24. Histogram showing the percentage of viral release in the absence (blue) and in the presence of 0.25 μM or 1 μM of latrunculin B (*N* = 5 independent experiments). Each exact *p*-values given in graphs were obtained using two-tailed Mann-Whitney tests. Data are presented as mean values +/- SEM. Source data are provided as Source data file.

concentrations measured by alphaLISA. To see if this effect comes from an increase in viral particle release (Western blot) or in total CAp24 release (alphaLISA), we quantified viral particle release from latrunculin B treated Jurkat T cells infected with the single round virus

using ultracentrifugation through a sucrose cushion followed by western blot analysis. This allowed us to analyze the release of entire viral particles, as well as the level of mature capsid protein and immature Gag precursor in these purified virions (Fig. 1C). Western

blot quantification showed a significant increase in particle release as well as in mature capsid protein and immature Gag(i)GFP release upon 0.25 μM of latrunculin B treatment (DMSO, 36 ± 3% versus LatB, 50 ± 4%) and with a higher concentration of 1 μM (DMSO: 37 ± 3% versus LatB: 69 ± 2%).

Our results showed that F-actin depolymerization by latrunculin B enhanced HIV-1 release from infected Jurkat T cells while F-actin stabilization by jasplakinolide tended to block particle release, suggesting that polymerized actin could act as a barrier for HVI-1 Gag assembly or particle release while F-actin depolymerization could have promoted this process.

### Actin debranching drug CK666 increases HIV-1 release from infected CD4+ T lymphocytes without impairing virus maturation

Since our results showed that actin depolymerization favored HIV-1 release, we explored the effect of reducing cortical cellular F-actin density using the pharmacological drug CK666 on HIV-1 infected host T cells. Previously, using transfection of molecular HIV-1 clones and siRNA-targeted cellular genes, we reported that the Rac1 dependent IRSp53/WAVE2/Arp2/3 signaling pathway triggering branched actin was involved in HIV-1 production in CD4+ T lymphocytes[16,27]. Here, we studied the effect of Arp2/3 inhibition by CK666 on HIV-1 release from infected T cells. CK666 is a pharmacological drug inhibitor of the Arp2/3 complex that stabilizes it into its inactive state by preventing Arp2 and Arp3 subunits from changing conformation[36]. Inactivation of Arp2/3 complex results in decreased branching of F-actin network[37], causing a decrease in F-actin intensity[22,38]. A range of CK666 concentrations (0 to 250 μM) was added to VSV-G pseudotyped HIV-1(i)GFPΔEnv infected Jurkat T lymphocytes (Fig. 2A). Quantification of HIV-1 p24 release by alpha-LISA showed a significant increase of 73 ± 6% in the presence of 75 μM of CK666 ($N = 2$ independent experiments, $n = 8$), this concentration being also the minimal effective non-toxic concentration. For this reason, this concentration was used for all further experiments. We monitored drug toxicity by measuring cell viability by cell Titer assay (Supplementary Fig. 3C), and no toxicity was detected with CK666 in both cell type. Flow cytometry revealed, in both conditions (with and without CK666), an equivalent percentage (16-18%) of Gag(i)GFP expressing cells (Supplementary Fig. 3B). As a negative control, the inactive form of the drug, CK689 showed no effect on viral release even at higher concentrations (Fig. 2B). We then followed the kinetics of HIV-1 release, every 12 hours for 4 days post infection on Jurkat T cells infected with VSV-G pseudotyped HIV-1(i)GFP-ΔEnv in the absence (DMSO) or in the presence of CK666. We calculated the half time ($t_{1/2}$) corresponding to the time needed to reach 50% of the maximum viral production. Results showed that HIV-1 p24 release was faster upon CK666 treatment, with a $t_{1/2}$ of 43 hours as compared to DMSO (48 hours), and a doubling time of 4.6 hours, 28% faster than the control DMSO (6.4 hours) (Fig. 2C). This increase in HIV-1 p24 release in the presence of CK666 was observed not only on infected Jurkat T cells (87 ± 7% increase, $N = 3$ independent experiments, $n = 8$) (Fig. 2D), but also on PBLs (37 ± 6% increase, $N = 5$ independent experiments, $n = 15$; 2 different donors), and also with the wild type virus (51 ± 4%, $N = 2$ independent experiments; $n = 6$; 2 different donors) (Fig. 2E). p24 ng/ml release raw values corresponding to the normalized data in Fig. 2E were also shown in Supplementary Fig. 4. Therefore, we questioned a possible effect of CK666 on Gag maturation. To test the effect of CK666 on HIV-1 Gag Pr55 release, we quantified, by immunoblots, both HIV-1 immature Gag Pr55 and mature HIV-1 p24 release (Fig. 2F, G). Our results revealed 26% increase in HIV-1 (p24 and Gag) release in the presence of CK666 as compared to the control DMSO in Jurkat T cells (DMSO: 53 ± 2% versus CK666: 67 ± 3%; $N = 6$ independent experiments,) (Fig. 2F). This increase was also observed in infected PBLs with single

round virus, with a significant 34% increase in viral release with CK666 as compared to the control (DMSO: 50 ± 5% versus CK666: 67 ± 2%, $N = 7$ independent experiments, 2 donors) (Fig. 2G), and confirmed with wild type HIV-1 infected primary lymphocytes, with an increase of 26% (DMSO: 53 ± 5% versus CK666: 67 ± 4%; $N = 10$ independent experiments) (Fig. 2F). In addition, we calculated the ratio of Gag maturation in virion (% of mature CAp24 / total Gag +CAp24) in infected Jurkat T cells as well as in infected primary lymphocytes and our data showed no significant change in Gag maturation upon CK666 treatment as compared to the control (DMSO) (Supplementary Fig. 5). Together, these results asserted that inhibition of Arp2/3 mediated branched actin by using CK666 increased HIV-1 release in infected T lymphocytes, in a Gag dependent manner, suggesting that the effect of CK666 occurred prior to viral particle maturation and therefore before budding. The increase in viral release upon actin debranching due to CK666 was in coherence with the restrictive role of F-actin in viral particle production as reported with latrunculin B treatment (Fig. 1). Together these results showed that less F-actin favors HIV-1 particle release and suggested a role of Arp2/3 inhibition in this process. Thus, we next explored the effect of Arp2/3 inhibition via CK666 on HIV-1 assembly at the plasma membrane of CD4+ T cells.

### Debranching actin increases the cell surface density of HIV-1 Gag clusters in infected T lymphocytes

For this purpose, we imaged F-actin and Gag at the plasma membrane of Jurkat T-cells using TIRF microscopy in a first attempt, supplemented by single-molecule localization STORM microscopy. By measuring changes in fluorescent phalloidin F-actin labeled intensities in Jurkat T cells, we first controlled that the decrease of F-actin content upon CK666 treatment was significant 48 h post infection (42% decrease of F-actin content from $(1.97 \pm 0.15).10^4$ a.u. in fluorescence intensity in DMSO to $(1.13 \pm 0.13).10^4$ a.u in CK666, $N = 15$ cells) (Fig. 3A). Flow cytometry analyzes of CK666 treated cells also confirmed this decrease (23% decrease in F-actin mean intensity) (Supplementary Fig. 4A). We then quantified the surface density of HIV-Gag(i)GFP platforms 48 h post infection at the cell membrane of infected Jurkat CD4 + T cells (Fig. 3B). Using classical TIRF microscopy we measured 0.21 ± 0.01 ($N = 9$ cells) platforms per μm² when the CD4 + T lymphocytes were treated with CK666, which was twice the surface density observed in the DMSO control (0.10 ± 0.01 platforms per μm², ($N = 9$ cells)) (Fig. 3B). However, due to low lateral resolution of TIRF microscopy, these platforms could have contained several distinct clusters of Gag, that would each represent a different state of assembly (from starting to fully assembled and possibly released particle). Therefore, we applied the precision localization efficiency of STORM microscopy with 22 nm resolution (see Supplementary Fig. 6B and Method section), to specifically identify individual HIV-1 Gag(i)GFP clusters, labeled with GFP nanobodies, that were absent in immuno-labeled non-infected cells (Fig. 3C). We thus quantified the Gag clusters mean diameter that results in 94 nm for DMSO- and 107 nm for CK666-treated infected cells (Fig. 3C), e.g., in the same range than HIV-1 infected T-cells that did not experience any treatment (mean=114 nm, Supplementary Fig. 6D). These average cluster diameters most probably reflected Gag assembling clusters since fully assembled viral particles being roughly 139 ± 7 nm in diameter[39]. Confirming TIRF observations, we noted a significant increase in individual cluster density (3.3 ± 0.1 clusters per μm², $N=5$ cells, $n = 50$ ROI (region of interest)) in cells treated with CK666 compared to control (2.8 ± 0.1, clusters per μm², N=5 cells, $n = 50$ ROI (region of interest)) (Fig. 3). With these results, we confirmed that inhibition of branched actin using CK666 increased HIV-1 Gag assembly cluster formation. These data suggested that the increase in viral release observed with CK666 (Fig. 2) was a consequence of increased viral assemblies occurring simultaneously at the plasma membrane.

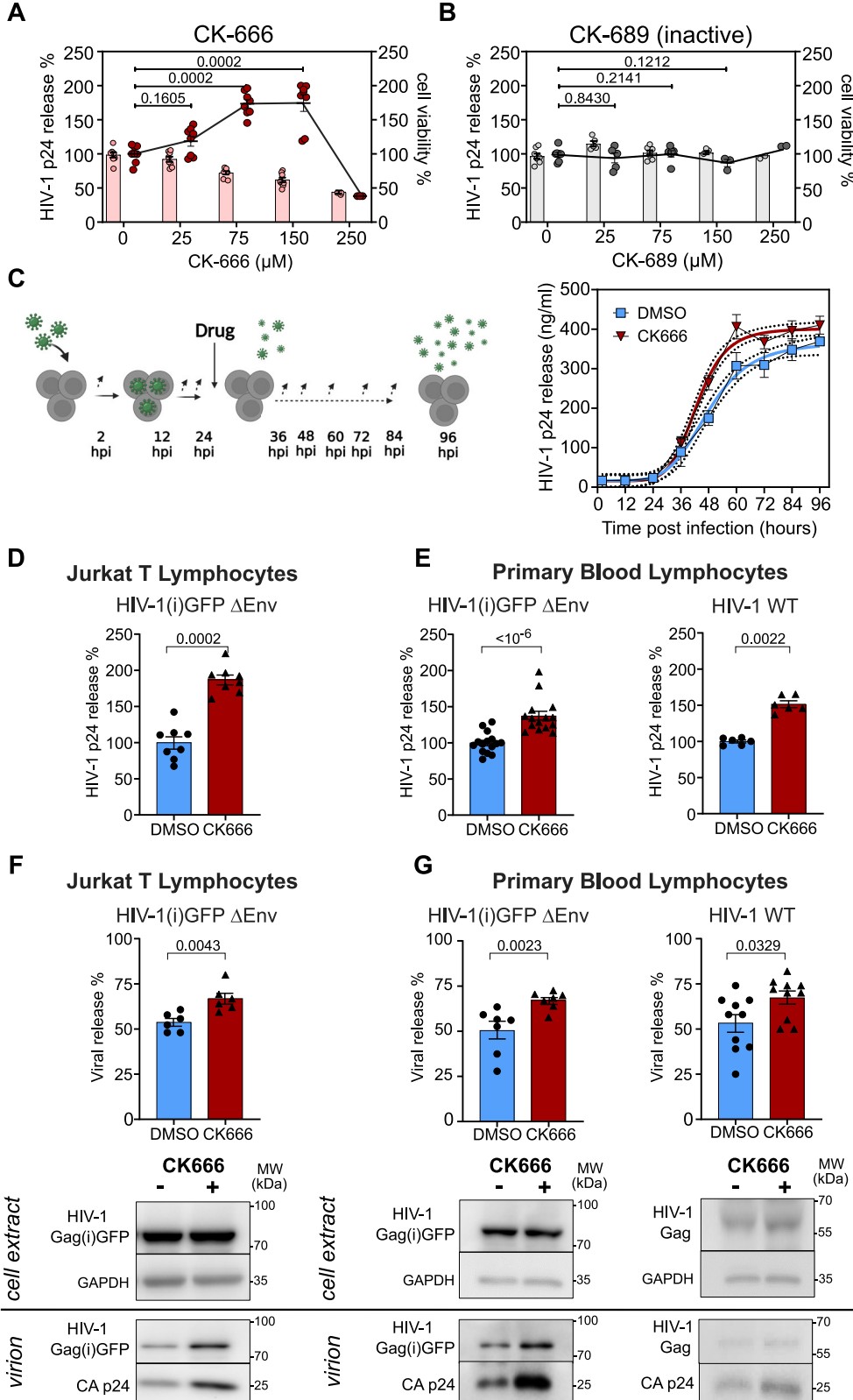

## F-actin density decreases at Gag assembly sites in infected CD4⁺ T lymphocytes

We next addressed the density of F-actin in the vicinity of HIV-1 Gag assembly sites at the surface of HIV-1 infected T cells, using TIRF microscopy coupled or not to STORM (Fig. 4).

TIRF images of dual labeled F-actin and Gag(i)GFP at the plasma membrane of pseudotyped single round HIV-1 infected Jurkat CD4⁺ T lymphocytes were used to quantify the F-actin intensity in regions of interest (ROI) enriched in Gag(i)GFP (Gag+ area) or not (Gag- area) (Fig. 4A). We normalized the F-actin intensity in each ROI to the total F-actin intensity of the cell to bypass the heterogeneity in F-actin intensity between cells. We plotted F-actin intensity distribution in the presence of CK666 and DMSO as a control, and we observed that F-actin mean intensity of Gag+ area $(0.90 \pm 0.04, n = 69)$, was not significantly

**Fig. 2 | CK666 targeting branched actin favors HIV-1 release from infected T lymphocytes without impairing virus maturation. A** Left Y axis represents the relative percentage of HIV-1 p24 release (dots) measured by alphaLISA from infected Jurkat T lymphocytes with VSV-G pseudotyped HIV-1(i)GFPΔEnv treated 24 hours post-infection with 0, 25, 75, 150, and 250 μM of CK666 (scatter dots in red, means linked). $N = 2$ independent experiments; $n = 8$ for 0, 25, 75, and 150 μM; $n = 3$ for 250 μM (n, replicates). **B** Same with the inactive drug form CK689 (dots in black). For both, the Right Y axis represents the percentage of cell viability (bars) normalized to the control (no drug) measured with 96AQ CellTiter. Cell viability (%) is normalized to the control (no drug). $N = 2$ independent experiments; $n = 8$ for 0, $n = 5$ for 25, $n = 7$ for 75, $n = 3$ for 150, $n = 2$ for 250 μM (n, replicates). **C** Left: scheme of the protocol used to measure infection kinetics displayed on the graph. This scheme was created under BioRender. Right: Red and blue curves refer respectively to the fits of p24 release values for 75 μM CK666 and DMSO (control), ($n = 4$ replicates). **D** Histograms showing the percentage of HIV-1 p24 release normalized, in the presence of 75 μM CK666 (in red) as compared to DMSO (in blue) from Jurkat

T lymphocytes infected with single round virus ($N = 3$ independent experiments, $n = 8$ replicates), **E** Same for primary blood lymphocytes (PBL) infected either with single round virus ($N = 5$ independent experiments, $n = 15$ replicates; 2 donors) or HIV-1 WT ($N = 2$ independent experiments; $n = 6$ replicates; 2 donors). **F** Histograms showing the relative percentage of viral release with 75 μM CK666 (in red) or DMSO control (in blue) from Jurkat T lymphocytes infected with a single round virus ($N = 6$ independent experiments). **G** Same for PBL infected either with a single round virus ($N = 7$ independent experiments, 2 donors) or with HIV-1 WT ($N = 10$ independent experiments, 2 donors). The percentage of virus release is measured from immunoblot quantification (see Methods). Anti-p24 immunoblots (on cell extracts and purified virions) show immature Gag(i)GFP ( ~ 82 kDa) in the case of the single round virus, and HIV-1 Gag ( ~ 55 kDa) in the case of the WT virus, and mature CAp24, as indicated. GAPDH is used as a loading control in the same blot. Each exact *p*-values given in the graphs was obtained using two-tailed Mann-Whitney tests. Data are presented as mean values +/- SEM. Source data are provided as Source data file.

---

different than that of Gag-area ($0.94 \pm 0.01$, $n = 80$) in the control cells (DMSO). However, this intensity was significantly lower ($0.81 \pm 0.02$, $n = 72$) in Gag+ area than in Gag− area ($1.00 \pm 0.03$, $n = 80$) in CK666 treated cells (Fig. 4B). These results suggested that HIV-1 Gag assembly sites overlapping with less F-actin upon actin branching inhibition. We then imaged both F-actin and HIV-1 Gag at the plasma membrane of infected Jurkat T cells using two colors STORM microscopy under TIRF incidence (Fig. 4C, D). Using DBScan, we first identified and isolated HIV-1 clusters at the cell surface to generate a new localization image. We then observed the F-actin localization in the proximity of these clusters. The average size of cortical actin meshwork is considered to be between 100 to 200 nm[40]. Therefore, using Coordinate Based Colocalization (CBC), we decided to quantify the colocalization of F-actin present in an area defined by a 300 nm radius around the center of HIV-1 Gag (Fig. 4E–G). The CBC values were distributed from -1 (anticorrelated, i.e. every time there is a Gag localization, the first F-actin localization will be further than 300 nm), to 1 (fully correlated, i.e. each Gag localization is at the same position then the actin localization). We defined the -1 to -0.5 interval to represent (partially) anticorrelated localizations and, symmetrically, the 0.5 to 1 interval to represent the (partially) correlated localizations. Figure 4E represents the CBC distribution in Jurkat T cells with intact actin, showing an equivalent percentage of actin localization correlated or anticorrelated with Gag cluster localizations in the control DMSO. On the opposite, when treated with CK666 (Fig. 4F), the distribution was shifted towards anticorrelated actin localization regarding Gag clusters. Figure 4G represents the cumulative CBC distribution comparing both. These results were of importance as, although the HIV cluster surface density increased in CK666-treated T-cells, this increase in density was associated with less dense F-actin. As we had shown previously that CK666 treatment induced a decrease in the mean intensity of F-actin fluorescence, the STORM microscopy data revealed that such a decrease was associated with an apparent increase in the F-actin meshwork size (Supplementary Fig. 7), which could be responsible for the change in the CBC distribution we quantified here.

To check that, we then mimicked F-actin meshwork density in vitro on model membranes (Figs. 5A, B) and tested its effect on Gag assembly initiation.

## HIV-1 Gag clustering on model membrane is favored in low-density F-actin meshwork

HIV-1 assembly initiation is a diffusion-reaction process strongly depending on membrane-bound diffusion and concentration of Gag[41]. In this perspective, cortical actin spatial distribution (size and surface density of actin meshes) is a key point as it can not only impact the membrane tension[4] but also the membrane lateral diffusion[42]. We addressed possible diffusion modulations of HIV-1 Gag membrane-bound molecules by cortical actin meshwork and

their consequences on viral assembly. We have previously shown that PI(4,5)P$_2$ diffusion is strongly correlated to HIV-1 Gag membrane-bound lateral diffusion[13,14,43]. Therefore, using classical spot-variation microscopy[44], we first established Atto647N-PI(4,5)P$_2$ fluorescent lipid analog diffusion laws in infected Jurkat T-cells treated or not with CK666 (Fig. 5A, C). Linear fits of these diffusion laws (Supplementary Fig. 8A) showed that Atto647N-P(4,5)IP$_2$ diffusion coefficient significantly increased from $1.2 \pm 0.1 \, \mu m^2.s^{-1}$ to $2.4 \pm 0.2 \, \mu m^2.s^{-1}$ in CK666 treated cells ($N = 3$ different samples, $n = 6$ cells) (Fig. 5C). Interestingly, we also observed drift of the linear fit intercept value (Fig. 5D) from negative values ($t_0 = -7.8 \pm 0.9$ ms), suggesting a meshwork impeded lateral diffusion[44], to a value close to zero ($t_0 = 1.6 \pm 0.6$ ms), suggesting a return to Brownian lateral diffusion, when cells are treated with CK666. These data showed that, in Jurkat T-cells, Atto647N-PI(4,5)P$_2$ diffusion is restricted by the presence of cortical branched actin meshwork.

Directly monitoring the effect of actin meshwork on membrane-bound Gag diffusion in cells using FCS diffusion laws is a more complex problem since Gag is a protein partitioning between membrane and cytosol (Fig. 5E). Indeed, we have recently shown that small deviations of the axial laser position from the cell membrane surface generate increased values of the apparent membrane-bound diffusion coefficient[43] that might lead to mis-interpretation. To circumvent that issue, we set up in vitro a cortical actin/plasma membrane minimal system adapted from[45] where we could control the z-position of the laser beam. We made supported lipid bilayers of chosen lipid composition (see Material and Methods section for details) in which we also introduced the Atto647N-PI(4,5)P$_2$ and head-biotinylated lipids. Actin filaments preliminary bound to biotinylated phalloidin can be bridged to the membrane via a streptavidin, which also, thanks to its four reactive domains, connects the actin filaments, mimicking the branching of the cortical actin network. To simulate different scenarios, from none to dense cortical actin network, we used different concentrations of biotinylated lipid anchor (from 0.1 to 1 mol%) and different concentrations of actin (from 0 to 4.4 μM), resulting in various meshwork surface density as seen in Fig. 5B. We first controlled that our actin/SLB in vitro system to mimic the cell plasma membrane and its underlying cortical actin meshwork. For this, using classical spot-variation microscopy[44] as in Jurkat T-cells, we established Atto647N-PI(4,5)P$_2$ diffusion laws as a function of actin concentration (Fig. 5B, C & Supplementary Fig. 8B). As in Jurkat T-cells, we observed significant decreases in Atto647N-PI(4,5)P$_2$ diffusion coefficient with increasing actin and biotin concentration (Fig. 5C) (D = $2.3 \pm 0.3 \, \mu m^2.s^{-1}$ without actin and $1.0 \pm 0.1 \, \mu m^2.s^{-1}$ with the densest actin network). Similarly to Jurkat T-cells, the intercept values of the Atto647N-PI(4,5)P$_2$ diffusion laws progressively and significantly changed from slightly positive values ($t_0 = 1.9 \pm 1.2$ ms) for SLB without actin meshwork to negative values ($t_0 = -4.1 \pm 0.7$ ms) in the highest actin and biotinylated

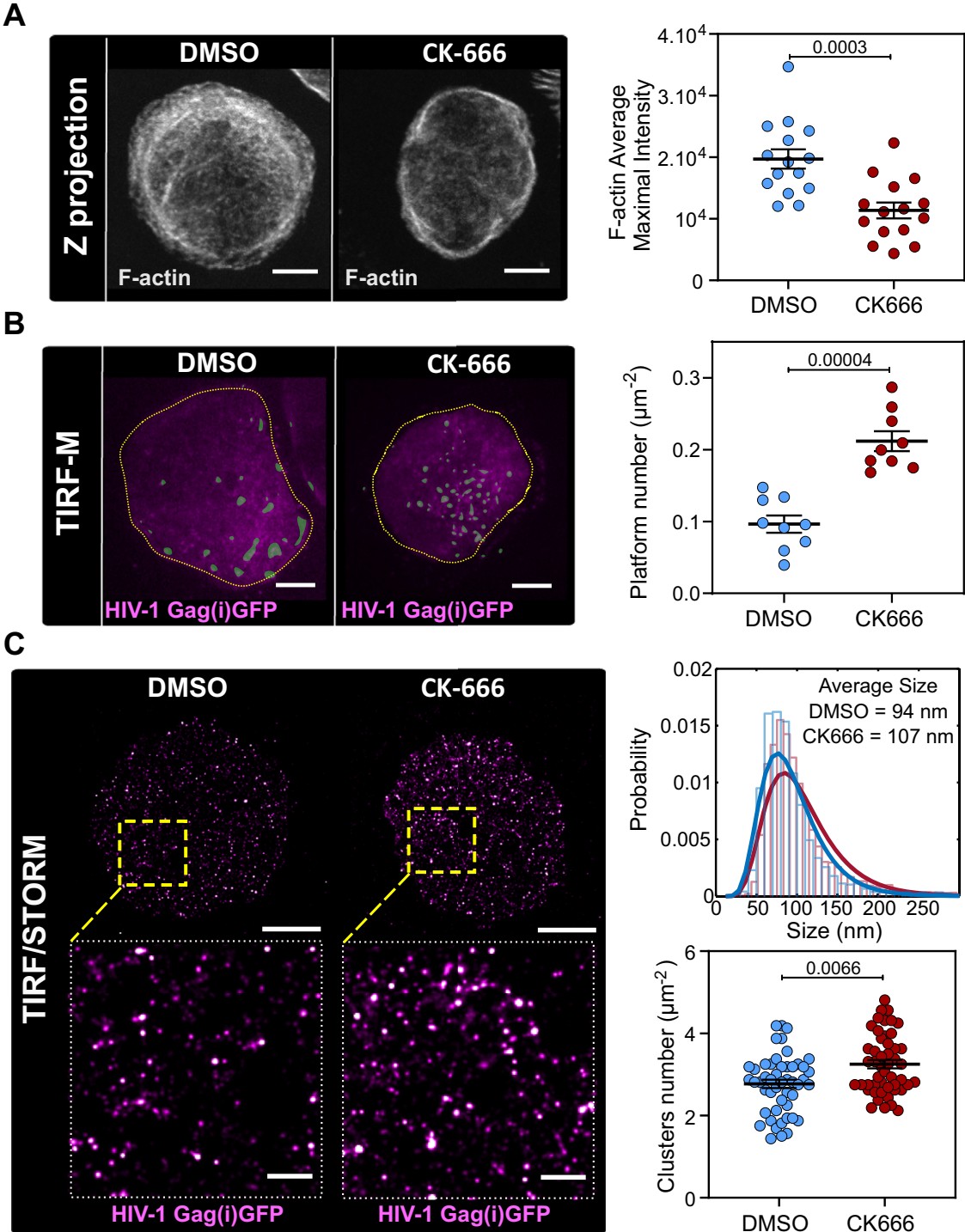

**Fig. 3 | Debranching F-actin increases HIV-1 assembly at the plasma membrane of infected CD4 T lymphocytes. A** Z projection of confocal images showing F-actin labeling in infected CD4+ Jurkat lymphocytes treated with DMSO and CK666. The scale bar is 5 μm. Dot plot of F-actin quantification in cell treated with DMSO or CK666, N = 15 cells. **B** TIRF-M images of Jurkat CD4 + T lymphocytes infected with VSV-G pseudotyped HIV-1(i)GFPΔEnv (magenta) and treated with 75 μM CK666 or not (DMSO) showing HIV-1 Gag assembly clusters at the cell surface. The scale bar is 5 μm. Dashed contouring showed the area of the cell used to calculate the density of Gag assembly platforms. Binary images of clusters indicated by superposed in faded green. Dot plot showing the number of HIV-1 Gag assembly platforms per

surface (μm⁻²), N = 9 cells. **C** STORM/TIRF images of HIV-1 Gag at the surface of Jurkat CD4 + T lymphocytes infected with VSV-G pseudotyped HIV-1(i)GFPΔEnv (in magenta) treated with 75 μM CK666 or not (control DMSO). The scale bar is 5 μm for full cell images, 1 μm for zoomed images. Right, upper graph: size distribution of HIV-1 Gag(i)GFP clusters quantified by DBScan (n = 4092 clusters for DMSO in blue, and n = 4484 for CK666 in red, N = 5 cells). Right, lower graph: dot plot showing the number of assembly clusters per surface (μm⁻²), N = 5 cells, n = 50 ROI (region of interest), Each exact p-values given in graphs were obtained using two-tailed Mann-Whitney tests. Data are presented as mean values +/- SEM. Source data are provided as Source data file.

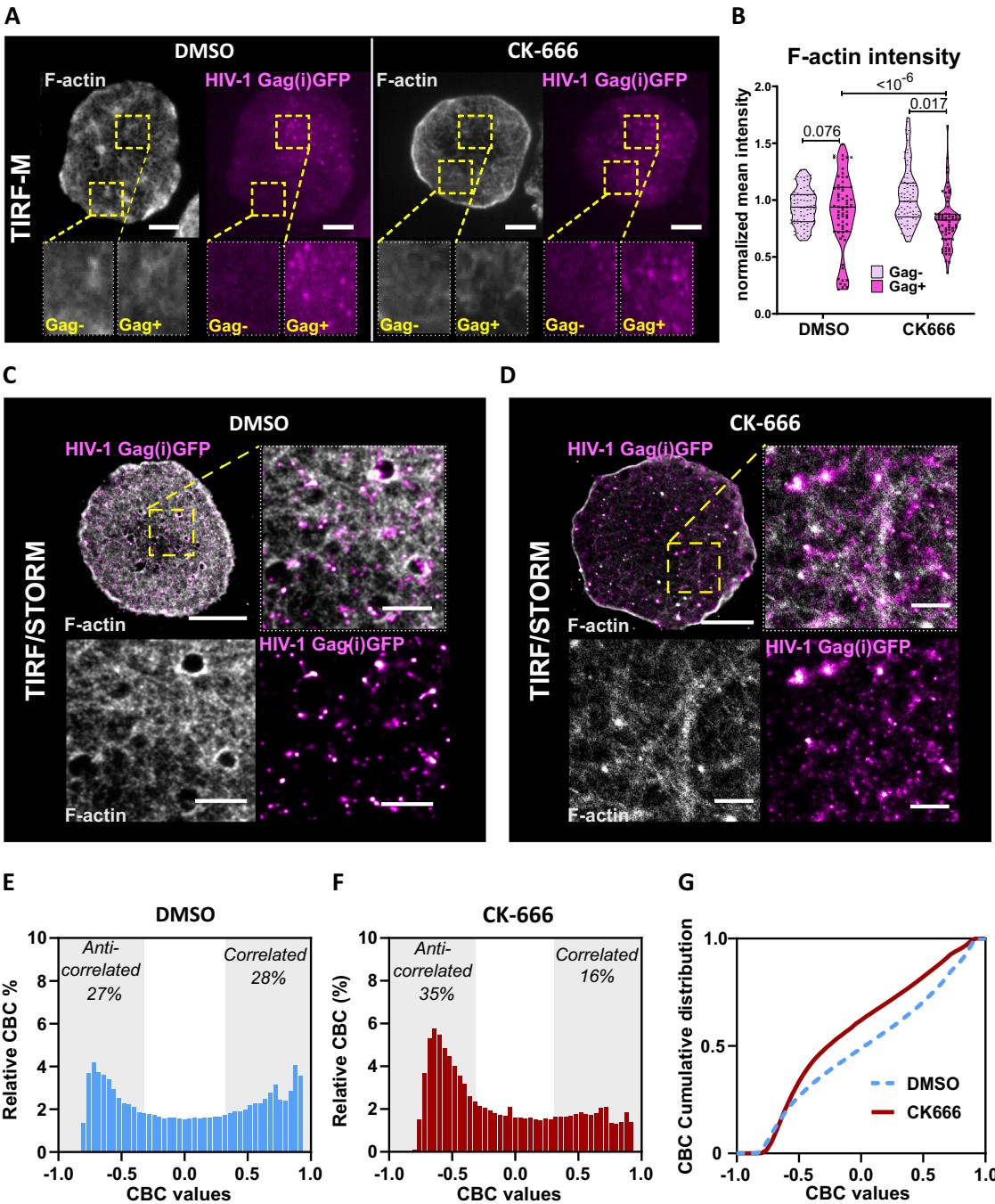

**Fig. 4 | Local decrease of F-actin density at the T cell surface is in favor of HIV-1 Gag assembly site localization. A** TIRF-M images of infected Jurkat T cells treated with DMSO (left panel) or CK666 (right panel), showing HIV-1 Gag(i)GFP assembly platforms (in magenta) with F-actin labeled with Phalloidin 647 Alexa Fluor (in gray). Zoomed images showing the region of interest (ROI) area enriched (Gag + ) or not (Gag-) with Gag assembly platforms. The scale bar is 5 µm. **B** Violin plot of F-actin intensity distribution in an area enriched of HIV-1 Gag (dark pink) and area with no HIV-1 Gag (clear pink) (N = 10 cells. 62≤n ≤79 different ROIs, data are presented as median value +/− 1st interquartile range (IQR)). STORM/TIRF images of infected Jurkat T cells treated with DMSO (in **C**) or CK666 (in **D**), labeled with nanobody Alexa Fluor 568, showing HIV-1 Gag(i)GFP individual assembly clusters (in magenta) with F-actin labeled with Phalloidin 647 Alexa Fluor (in gray). The scale bar is 5 µm for full-cell images. The scale bar of zoomed images is 1 µm. CBC distribution shows the % of HIV-1 Gag / F-actin correlation ( > 0.5) and anticorrelation (<−0.5) in DMSO (in **E**) and CK666 (in **F**). **G** Cumulative distribution of CBC values for DMSO (dashed blue curve) and CK666 (red curve). Each exact *p*-value given in the graphs was obtained from two-tailed Mann−Whitney test. Source data are provided as Source data files.

lipid concentrations. Then, we introduced fluorescently labeled recombinant viral Gag proteins in the bulk and, after 20 min of incubation, we performed spot-variation FCS experiments and analyzed the diffusion laws as in[43] to monitor the variation of HIV-1 Gag monomers membrane diffusion coefficients and their membrane partitioning in the different actin meshes (Fig. 5E−G, Supplementary Fig. 9A) To

avoid any self-assembly of Gag molecules on the surface of the SLB, that would have immediately modified the membrane diffusion coefficient ($D_{bound}$) as well as the estimated partition coefficient (Kp = membrane Bound/ bulk Free Gag), we have tuned Gag concentrations to be sufficiently low (10 nM)[13,43]. We observed a significant (4x) decrease of membrane Gag diffusion in the presence of the densest

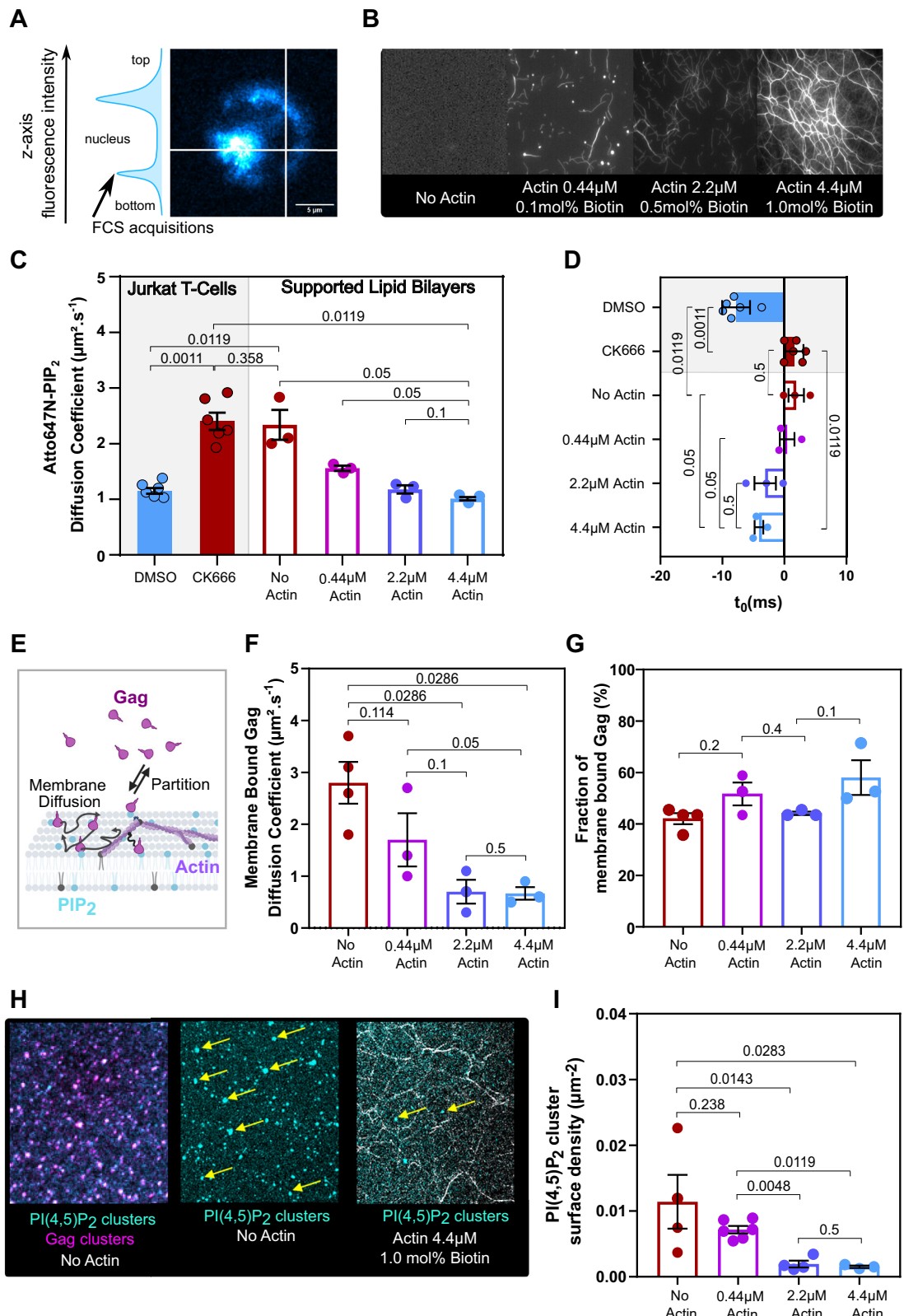

actin network ($D_{bound} = 0.7 \pm 0.1\ \mu m^2.s^{-1}$) compared to the free membrane ($D_{bound} = 2.8 \pm 0.4\ \mu m^2.s^{-1}$) (Fig. 5F). Interestingly, the dense actin network did not decrease the apparent partition coefficient of HIV-1 Gag (Supplementary Fig. 9A) since we observed no significant change in the fraction of membrane-bound Gag (Fig. 5G) with successive increasing concentration of actin and biotinylated lipid, varying between $42 \pm 2\%$ in the absence of actin and $58 \pm 7\%$ for the densest

actin network. A constant partitioning between membrane and cytosol, independently of the actin meshwork surface density, was also observed using membrane flotation assays[16,39] in HIV-1 infected Jurkat T-cells, treated or not by CK666 (Supplementary Fig. 9B, C). These data showed that the actin network strongly decreased the lateral diffusion of lipid membrane-bound HIV-1 Gag proteins while their concentration remained constant or slightly increased with the densest network.

**Fig. 5 | Dense actin meshwork decreases PI(4,5)P₂ and HIV-1 Gag diffusion and Gag self-assembly on model membranes. A** Spot variation FCS measurements were performed at the bottom of Jurkat T-cells labeled with Atto647N-PI(4,5)P₂ fluorescent lipid analogs, to monitor its motion, using fluorescence intensity z-scan to retrieve the plasma membrane. **B** Images of increasing F-actin meshwork density at the surface of supported lipid bilayers (SLB) with increasing biotinylated lipid and actin concentrations. **C** Atto647N-PI(4,5)P₂ diffusion coefficient in Jurkat T-cells treated or not with CK666 (left) and in SLB with increasing biotinylated lipid and actin concentrations ($N$ = 3 independent experiments, with $n$ = 6 for cells, $N$ = 3 independent experiments with $n$ = 3 for SLBs, one-tailed Mann-Whitney was used to test significant decrease of diffusion coefficient among the different conditions). **D** y-intercept values of the sv-FCS diffusion law linear fit ($t_0$) for the different conditions in **C**. **E** Scheme illustrating Gag motions in the vicinity of a membrane. The binding/unbinding equilibrium gives Gag partitioning in addition to lateral diffusion of the membrane-bound Gag. This scheme was created under BioRender

(full license to DM). **F** Membrane bound diffusion coefficient of Gag added to the different SLBs described in B) ($3 \leq N \leq 4$ different SLBs for each condition, one-tailed Mann–Withney to test decrease of diffusion coefficient with increasing actin concentration). **G** Fraction of Gag bound to the membrane for the different SLBs ($3 \leq N \leq 4$ different SLBs for each condition, two-tailed Mann–Withney to test for differences in the mean binding with increasing actin concentration). **H** Atto647N-PI(4,5)P₂ and HIV-1 Gag-A488 clusters were imaged at the surface of SLB 20 min after Gag injection (left), Atto647N-PI(4,5)P₂ clusters were imaged 20 min after unlabeled-Gag injection on SLBs without actin (center) and with 4.4 μM actin (right). **I** Quantification of clusters surface density in SLBs with increasing density of actin meshwork ($3 \leq N \leq 4$ SLBs for the different conditions, and $3 \leq n \leq 6$ for the different areas imaged in the different SLBs, one-tailed Mann–Whitney was used to test the decrease of Atto647N-PI(4,5)P2 clusters induced by addition of Gag with increasing actin concentration). Exact *p*-values are given in graphs. Data are presented as mean values ± SEM. Source data are provided as Source data files.

We, therefore, questioned whether this diffusion restriction could have effectively played a role in the assembly process as suggested[41]. For this, we incubated the cortical actin model membranes with a higher concentration of Gag (250 nM), for which self-assembly has been shown to occur[13]. Then, after 20 min of Gag incubation, we imaged and quantified the Atto647N-PI(4,5)P₂ clusters generated by Gag self-assembly (Fig. 5H and Supplementary videos S1 for Atto647N-PI(4,5)P₂, S2 for Gag and S3 for merge Atto647N-PI(4,5)P₂ and Gag), presented at the surface of the SLB either in the absence or in the presence of the actin mesh (Fig. 5H, I). Indeed, we previously showed that recombinant purified HIV-1 Gag protein induces PI(4,5)P₂ cluster formation during assembly on this lipid membrane composition[13] (See also Supplementary Video S1) and that this fluorescent PI(4,5)P₂ clusters colocalized with fluorescently-labeled Gag clusters (Supplementary Video S3). Interestingly we observed a constant decrease of clusters with increasing surface density of actin meshes. Quantification showed that the surface density of PI(4,5)P₂ clusters induced by Gag in the absence of actin was 10 times higher than in the presence of the densest actin network, where Gag membrane diffusion was strongly decreased (Fig. 5F) although the membrane-bound Gag concentration was slightly (not significantly) increased.

Therefore, these results supported that lateral diffusion of monomers was a key factor governing the decrease of clustering with increasing actin network surface density. Our model membranes showed that the actin meshwork influenced HIV-1 Gag membrane diffusion and Gag-dependent PI(4,5)P₂ clustering, with no effect on Gag membrane, suggesting that F-actin surface density played a major role in the initiation of Gag assembly process and that a decrease in F-actin surface density would favor Gag assembly.

## Gag recruits the actin factor Arpin to favor HIV-1 assembly and particle release

In cells, the CK666 drug is mimicking the inhibition of branched F-actin, showing that actin meshwork with a decrease in branching was more favorable to Gag assembly cluster formation, resulting in an increase in particle release in HIV-1 infected T lymphocytes. We thus assessed the potential role of a cellular host factor that would prevent actin branching via Arp2/3 inhibition, since regulation or sequestration of this factor by Gag could indeed locally modify the degree of actin branching at on-going HIV-1 assembly sites. Arpin was a good candidate as one of the newly discovered inhibitors of Arp2/3[46]. Indeed, Arpin contains a carboxy-terminal acidic Arp2/3-binding motifs that blocks Arp2/3 complex into its inactive state by binding to the hydrophobic cleft of Apr3 and inhibiting the complex conformation change[47] in a manner that is similar to the complex change inhibition used by CK666[36]. We therefore examined the involvement of Arpin in HIV-1 production in CD4⁺ Jurkat T lymphocytes. Arpin Knock-down using siRNA in pseudotyped single round virus infected CD4⁺ T lymphocytes was tested on virus release using western blot (Fig. 6A). The

result showed that Arpin is necessary to HIV-1 production in T cells: with 44 ± 4% inhibition of Arpin (Fig. 6B), a significant decrease of HIV-1 release from 63 ± 4% in the control to 47 ± 3% in the presence of siArpin was observed ($N$ = 3 independent experiments) (Fig. 6C). Normalized to the control, viral release in cells treated with siArpin decreased by 25 ± 6% (Fig. 6D). Furthermore, to test whether Arpin induced HIV-1 release decrease necessitate its recruitment at the cell membranes, we performed a cell fractionation assay followed by immunoblots and we calculated Arpin cell membrane binding in HIV-1 infected compared to non-infected CD4⁺ T lymphocytes (Fig. 7A, B). Results showed only 3.7 ± 0.4% ($N$ = 3 independent experiments) of Arpin binding to cell membranes in non-infected cells, while this binding increased significantly with HIV-1 infection, with up to 8.5 ± 1.5% ($N$ = 3 independent experiments) of Arpin membrane binding (Fig. 7C). For a sake of comparison, the ESCRT-I machinery protein, Tsg101, known to interact with the p6 domain of Gag for efficient particle budding[48] was taken as a positive control. We also detected an increase in Tsg101 membrane binding from 29 ± 2% ($N$ = 3 independent experiments) in non-infected cells to 45 ± 3% ($N$ = 3 independent experiments) in infected cells (Fig. 7C). To confirm the increase in Arpin membrane binding upon HIV-1 infected T cells, we performed STORM imaging of Arpin, using the same anti-Arpin antibody used in western blotting and in TIRF-Microscopy. We compared non-infected Jurkat T cells with single round HIV-1(i)GFP infected T cells and quantified a significant 10-fold increase of membrane-bound Arpin localization surface density upon HIV-1 infection, from 62 ± 7 localizations per μm² in non-infected cells to 626 ± 65 localizations per μm² in infected cells ($n$ = 100 different areas of 1 μm²) (Fig. 7D).

Finally, we looked for a possible interaction between Arpin and HIV-1 Gag. For this we performed an immunoprecipitation assay, using 293 T HEK model cell line expressing Gag only (Fig. 7E). Indeed, only cell extract incubated with polyclonal anti-Arpin antibodies revealed a band of HIV-1 Gag by immunoblots and no band was seen with either no antibody or non-specific rabbit serum as negative control (Fig. 7E). These results reinforced the existence of an association between HIV-1 Gag and Arpin, suggesting that Gag diverted Arpin to the cell membrane. Altogether these results show that Arpin was involved in HIV-1 particle production. By capturing Arpin at or bringing it to the assembly site, Gag could locally enhanced actin debranching which in turn favored HIV-1 production as it was with the actin debranching drug CK666.

## Discussion

The findings of this study have revealed a role for branched F-actin during Gag assembly in HIV-1 infected host CD4 T lymphocytes. It has been long-held debate about whether F-actin might have played a role in HIV-1 assembly and particle release in various cell lines[21]. Some, using TIRF microscopy, reported that HIV-1 Gag did not use F-actin, as drugs have no effect on Gag assembly kinetics in adherent cells[49].

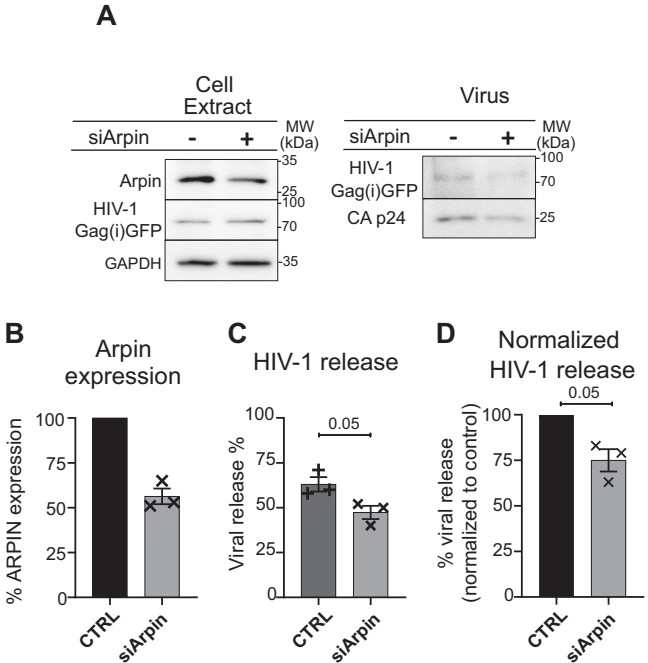

**Fig. 6 | Arpin is involved in HIV-1 particle release from infected Jurkat T lymphocytes. A** Anti-Arpin and anti-p24 immunoblots of VSV-G pseudotyped HIV-1(i) GFPΔEnv infected Jurkat CD4⁺ T lymphocytes cell extracts with siArpin (+) or siControl (-) and the corresponding purified released virus. GAPDH is used as a loading control. **B** Histogram showing the percentage (%) of Arpin gene expression after siRNA treatment corresponding to the Western blot in **A**) ($N = 3$ independent experiments). **C** Histogram showing the percentage of viral release calculated from HIV-1 Gag(i)GFP and HIV-1 CAp24 release observed in the Western blot in **A**), $N = 3$ independent experiments, one-tailed Mann Whitney to test for decrease in particle release. **D** Histogram showing the decrease in viral release (25 ± 3%) upon siArpin treatment normalized to the control (CTRL), $N = 3$ independent experiments, one-tailed Mann–Whitney to test for the decrease in particle release. Data are presented as mean values +/- SEM. Exact p-values are given in graphs. Source data are provided as Source data files.

stabilize actin filaments, had a dramatic effect, dose-dependent inhibition, on virus release (Fig. 1 and Supplementary Fig. 2). Overall our results indicated that F-actin dynamic tending towards depolymerization was favorable during HIV-1 assembly and particle release. Furthermore, the role of Arp2/3-mediated branched actin in HIV-1 assembly and release was unexplored. We thus studied the effect of a direct Arp2/3 inhibition on HIV-1 release from infected both primary T lymphocytes and Jurkat T cell line either using the CK666 drug or by targeting Arpin using siRNA. CK666 is a drug inhibitor of Arp2/3 complex that stabilizes it into its inactive state[36], resulting in decreased branching of F-actin network[37]. Importantly, we showed here that Arp2/3 inhibition by CK666 increased HIV-1 particle release from infected Jurkat or primary T lymphocytes and that it was coupled with an acceleration of the release without affecting virus maturation (Fig.2). This result prompted us to set up an infectious in vitro system to look at the level of virus assembly at the cell surface of VSV-G pseudotyped HIV-1(i)GFPΔEnv infected T cells using super-resolution STORM/TIRF-Microscopy (Fig.3), a tool of choice to look at HIV-1 particle assembly and budding[57–59] and cortical actin meshwork[60,61] at the cell plasma membrane. Interestingly, we detected an increase in HIV-1 Gag assembly clusters surface density at the plasma membrane of infected T lymphocytes along with a decrease in F-actin meshwork density at the cell surface. Thanks to super-resolution microscopy, we could distinguish individual HIV-1 Gag-labeled clusters on the cell surface, and observed that the spatial density of these individual assembly clusters increased when F-actin meshwork was absent (due to branched actin inhibition) (Fig. 4). Indeed, Arp2/3 inactivation has been shown to increase the actin meshwork size[37] and lipid diffusion have been shown to be remarkably faster in cell treated with CK666[62]. Knowing that there is a strong interplay between HIV-1 Gag and the phospholipid PI(4,5)P₂ at the plasma membrane inner leaflet during HIV-1 assembly[11,12,14] and that HIV-1 Gag membrane diffusion has been proposed to be important at the very first step of assembly, i.e. Gag nucleation on membrane[13,41], we explored the consequences of diffusion restriction by the actin meshwork on HIV-1 Gag self-assembly in vitro (Fig. 5). Using controlled molecular composition model systems, we show that HIV-1 Gag diffusion impeding, observed with increasing density of branched actin, has a drastic impact on Gag self-assembly initiation (Fig. 5). Although the use of minimal in vitro system does not imply that it directly reflects the situation in cells, this result is remarkably in line with what we observe in infected cells (Fig. 4). Furthermore, debranching actin with CK666 had no effect either on Gag or on Tsg101 cell membrane binding (Supplementary Fig. 8) supporting that debranching actin effect occurred at the level of Gag assembly but not viral budding. Although we have not addressed that effect, there is a correlation between the activation level of Arp2/3 and the membrane tension[63,64]. Inhibition of Arp2/3 decreases membrane tension, and it has been theoretically shown that lower membrane tension favors the assembly and budding of enveloped viruses[65]. This could therefore be an additional effect to what we observe in favor of HIV-1 assembly and release upon actin debranching.

Taken together, we propose that upon HIV-1 infection in T cells, the retroviral Gag reaches the cell plasma membrane for assembling in already-existing area less dense in cortical F-actin and that thereafter Gag could recruit the host cell factor Arpin is fostering this process since decrease in actin branching favors HIV-1 Gag assembling. This can either happen randomly, i.e. Gag reaches a site where actin branching is lower and the assembly is easier at this location, or this could be induced by Gag recruiting a debranching factor to favor the assembly nucleation at that membrane location. Since CK666 is a debranched actin drug targeting directly the Arp2/3 complex, we hypothesis that Gag could hijack the Arp2/3 inhibitor Arpin to promote HIV-1 Gag assembly locally at the cell membrane as Arpin has been studied for the similarity of its function with the effect of CK666[66,67]. Indeed, we found that Arpin membrane localization increased upon

Whereas others observed by cryo-EM the presence of branched actin and actin filaments, or sometimes its absence, underneath HIV-1 buds[15,50], reinforcing possible functional roles of F-actin in particle assembly. Other studies showed a role for actin regulating cofactors, such as filamin A, required for HIV-1 release[51] or that the Rac1-dependent IRSp53/Wave2/Arp2/3 signaling pathway was required for HIV-1 particle production[16] in which the membrane curvature factor IRSp53 fosters HIV-1 Gag assembly at the host cell membrane[16,27]. Several recent studies rely on the use of drugs perturbating F-actin polymerization[19,52,53]. Rac1 or CDC42 are both regulators of F-actin and IRSp53, at the crossed road of lamellipodia or filopodia generation[54,55] which itself depends upon Arp2/3 regulation. Here, we showed that Rac1 or CDC42 inhibitors decreased HIV-1 particle production in primary T lymphocytes (Supplementary Fig. 1). This is in concordance with a recent study showing that Arp2/3 mediated by CDC42 is involved in HIV-1 particle budding at the tip of filopodia in promonocytic cell line[35]. Latrunculin B treatment, known to reduce actin polymerization, has been reported in HIV-1 infected Jurkat T cells to have different effects, which can be the opposite, depending on the drug concentration and cell types[56] inducing either a decrease or increase in HIV-1 particle release which makes it difficult to conclude on the role of actin polymerization on HIV-1 release. Similarly, our results showed a decrease in HIV-1 production from primary T lymphocytes treated with latrunculin B (Supplementary Fig. 1) and an increase in the case of Jurkat T cells (Fig. 1). Here, we also showed that jasplakinolide, a drug known to block F-actin depolymerization and to

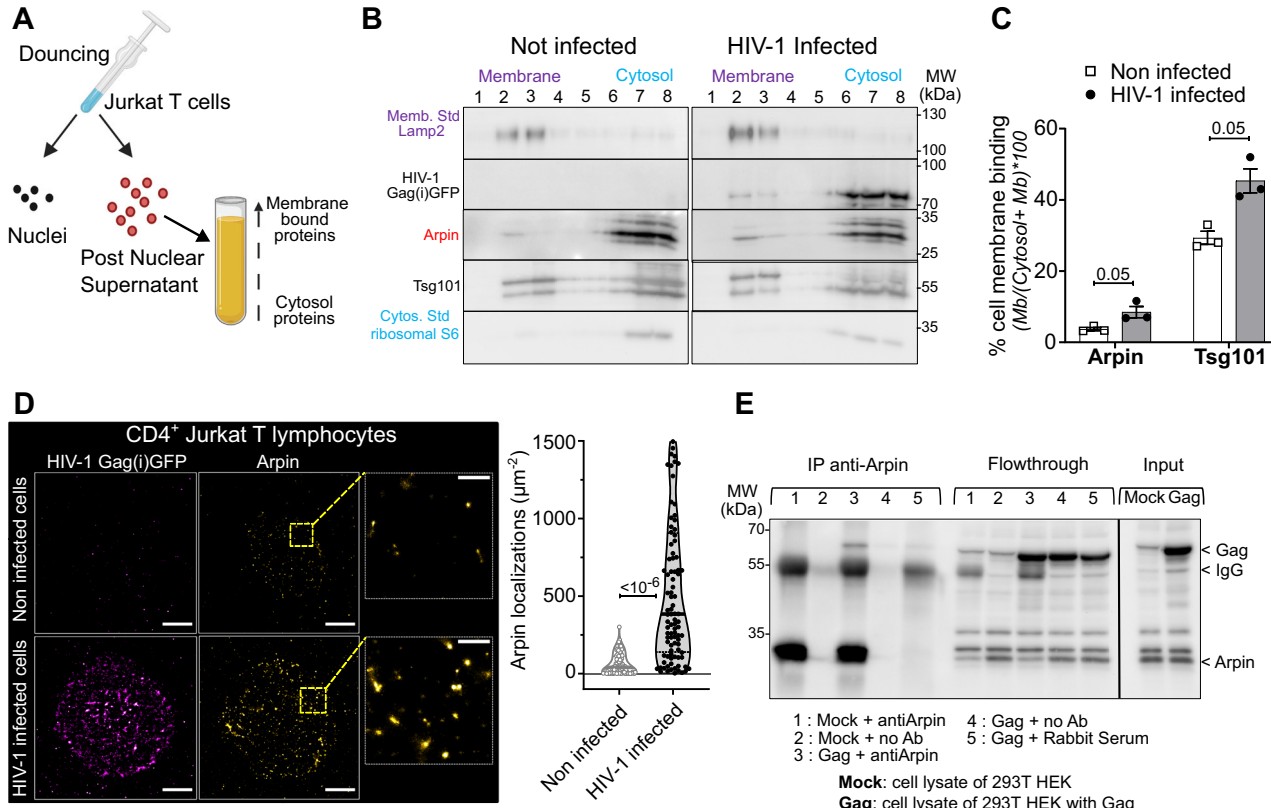

**Fig. 7 | Arpin binding to cell membranes increases in HIV-1 infected Jurkat T lymphocytes. A** Scheme showing the protocol of membrane flotation assay performed on infected Jurkat T cells (with the help of BioRender). **B** Immunoblots showing HIV-1 Gag(i)GFP, Arpin, and Tsg101 in the membrane (# 1, 2, and 3) and cytosolic (# 6, 7, and 8) fractions. The lysosomal Lamp2 protein was used as a membrane marker and the ribosomal S6 protein as a cytosol marker. **C** Histogram showing the % of Arpin, and Tsg101, binding to cell membranes in infected (in gray) or non-infected (in white) cells, $N = 3$ independent experiments, one-tailed Mann Whitney to test increase in membrane binding. **D** STORM/TIRF images of non-infected (upper panel) and VSV-G pseudotyped HIV-1(i)GFPΔEnv infected (bottom panel) Jurkat T cells showing, at the cell surface, HIV-1(i)GFP Gag labeled with nanobody Alexa-Fluor-568 (in magenta) and Arpin labeled with anti-Arpin and anti-rabbit secondary antibody ATTO-647N (in yellow). The scale bar is 5 μm for full-cell images. The scale bar is 1 μm for zoomed images. The right graph shows the

quantification of Arpin localization densities at the cell surface of non-infected (grey violin, each white dot corresponds to the number of localizations in a 1 μm² square ROI, $n = 100$ ROI) and of infected Jurkat T cells (black violin, each black dot corresponds to the number of localizations in a 1 μm² square ROI, $n = 100$ ROI). Two-tailed Mann-Whitney tests were used for localization density comparison. **E** Immunoblots showing the immunoprecipitation of HIV-1 Gag with the anti-Arpin antibody in transfected 293THEK model cell line. 1: non-transfected cell lysate incubated with anti-Arpin. 2: non-transfected cell lysate incubated with no antibody. 3: transfected cell lysate incubated with anti-Arpin. 4: transfected cell lysate incubated with no anti-Arpin. 5: transfected cell lysate incubated with rabbit serum. Bands showing HIV-1 Gag as well as antibody IgG (Ab) are indicated. $N = 3$ independent experiments. Data are presented as mean values +/- SEM, except in panel **D**) where data are presented as median value +/- 1st interquartile range (IQR). Exact p-values are given in graphs. Source data are provided as Source data file.

HIV-1 Gag expression in cells suggesting an interaction between Gag and Arpin that we revealed by an Arpin /Gag co-immunoprecipitation (Fig. 7). In addition, partial knock-down of Arpin significantly decreased HIV-1 particle production in infected CD4⁺ lymphocytes (Fig. 6A–D). This is reinforced by the fact that Arpin is recruited at the cell membrane upon HIV-1 infection (Fig. 7). These results strongly support the idea of HIV-1 Gag could recruit Arpin to prevent F-actin branching locally at the level of assembly sites when required.

We previously identified, together with IRSp53, Wave2, and Arp3 to be involved in HIV-1 particle production, using siRNA, under the regulation of Rac1 in CD4⁺ T cells[16,27]. At the cell plasma membrane, Rac1 was recently reported to drive a paradox regulation, named "incoherent feedforward loop"[68], where Rac1 can activate Wave2, the branching activator, but also Arpin, the branching deactivator of the Arp2/3 complex. Arpin activation is the key for promoting the feedback loop regulation of actin branching at the cell membrane, as reported most recently in cell migration[68] and here having an impact on HIV-1 production (Fig.6). This supports a need for HIV-1 Gag to control F-actin (de)polymerization and branching, most probably via Arpin and possible other cell factors, in a fine tune way, to promote HIV-1 assembly and particle release in the host CD4 T lymphocytes.

## Methods

### Ethics statement

Our research complies with all relevant ethical regulations in agreement with EFS OCPM, Montpellier, France (for laboratory research only under the convention #21PLER2019-0106, CNRS reference #208847). Informed consent was obtained for blood donation under the ethical regulations of the Etablissement du Sang Français (EFS) in Occitanie (OCPM), France.

### Cells culture

Blood from two different healthy donors was sourced from Etablissement Français du Sang, Montpellier (EFS OCPM). Primary Blood Lymphocytes (PBLs) were purified using a standard Ficoll gradient. PBLs were activated with phytohemagglutinin, PHA, (2 μg/ml), and interleukin 2, IL-2, (20 U/ml) for 72 hours before infection. PBLs, as well as human Jurkat T lymphocytes (human T-cell leukemia cell line) (ATCC-CRL-2899TM), were cultured in RPMI 1640 plus Glutamax (GIBCO) supplemented with 10% fetal bovin serum (FBS, Dominique Dutscher), sodium pyruvate and antibiotics (penicillin-streptomycin). Human embryonic kidney cells (293 THEK cell) (HEK 293T-ATCC-CRL-1575TM) were cultured in Dulbecco's Modified Eagle's Medium (DMEM, GIBCO)

supplemented with 10% fetal bovin serum (FBS, Dominique Dutscher), sodium pyruvate and antibiotics (penicillin-streptomycin). Cells were grown at 37˚C in a 5% CO2 atmosphere. All cell lines were provided and identified by The Global Bioresource Center ATCC - https://www.atcc.org/

## DNA plasmid and siRNA
The plasmid expressing Env-deleted HIV-1(i)GFP named pHIV-1(i) GFPΔEnv (NL4.3 strain) is from NIH (Catalog Number: ARP-12455, Lot Number: 140409). The plasmid expressing VSV-G named pVSV-G is from Addgene (plasmid #138479). The plasmid expressing HIV-1 codon-optimized Gag (pGag(myc), named pGag), and the plasmid expressing full wild-type HIV-1 (pNL4.3 WT) were described previously[14,16]. Stealth siRNA targeting Arpin gene (Arpin C15orf38, # HSS13447), as well as siRNA control (# 452002), were purchased from Invitrogen.

## Drugs and antibodies
The drugs used in this study are: CDC42 III inhibitor (Calbiochem), Rac1 inhibitor EHT1864 (Sigma Aldrich), Latrunculin B (Calbiochem), Jasplakinolide (Calbiochem), CK666 (Sigma Aldrich) and CK689 (Sigma Aldrich), DMSO (Calbiochem). The antibodies used in this study are: mouse antiCAp24 (6521 NIH aids reagent program), anti-Tsg101 antibody [EPR7130(B)] (Abcam ab125011), rabbit polyclonal anti-Arpin (Sigma ABT251) and rabbit polyclonal anti-Arpin (Invitrogen PA5-98574), anti-GAPDH coupled to HRP (horseradish peroxidase) (Sigma G9295). GFP-Booster Alexa Fluor 568 nanobody (gb2AF568) was purchased from Chromotek.

## In vitro model membranes
LUVs are prepared from a molar lipid mixture made of 69-x-y% Egg-PC, 28% Brain-PS, 2% Brain-PI(4,5)P$_2$, x% DSPE-PEG(2000)-Biotin (mol%, x vary from 0.1 to 1) and y% Atto647N-PI(4,5)P$_2$ (mol%, y = 0.1 for cluster imaging, y = 0.02 for FCS diffusion laws). All lipids purchased from Avanti Polar Lipids, Inc except Atto647N-PI(4,5)P2 which is a gift from Pr. C. Eggeling (IPHT, Jena, Germany). The lipids mixtures (0.5 mL at 1 mg/mL in chloroform) are evaporated for 20 min in a rotary evaporator and dried 10 min in a desiccator. The lipid film is rehydrated in 0.5 mL of filtered Na Citrate Buffer (Na Citrate Carlo Erba Reagenti (10 mM), NaCl (100 mM), EGTA (0.5 mM) pH4.6). Five consecutive times, the mixture is then frozen for 30 seconds in liquid nitrogen, heated back at 37 °C for 30 seconds, and vortexed for 30 seconds. To prepare the Supported Lipid Bilayer (SLB), 100 μL from the mix solution is diluted 1:5 in Na Citrate Buffer to reach a 0.2 mg/mL final concentration in a final volume of 500 μL. Mix solution is extruded 19 times with Avanti Polar Lipids Extruder, using 100 nm Nucleopore® Track-Etched Membranes, then sonicated 16 minutes (VWR Ultrasonic Cleaner USC-T) to obtain 30 nm extruded Small Unilamellar Vesicles (SUVs). Then, glass coverslips (VWR 25 mm Ø Cover Glasses, Thickness No. 1.5) are treated for 30 minutes with ozone in Ossila UV Ozone Cleaner and rinsed thoroughly with ultrapure water. The sample is delimited by a plastic cylinder of 7 mm diameter stuck to glass coverslips with Twinsil® (Picodent). 100 μL of 0.2 mg/mL SUVs solution is coated on the cleaned coverslip and incubated 40 minutes at 37 °C. The formed SLB is washed four times by: adding 100 μL of filtered TRIS HCl Buffer; Trizma® Base T1503 Sigma (10 mM), NaCl (150 mM), pH 7.4, pipetting carefully up and down seven times in the 200 μL solution to remove eventual vesicles attached to the Bilayer, and removing 100 μL. The final volume of buffer over the SLB is 100 μL.

## Actin polymerization and fixation on SLB
Rabbit skeletal muscle actin (Cytoskeleton, Inc) is resuspended at 10 mg/mL (232 μM), aliquoted and snap-frozen in liquid nitrogen. 1 μL actin is diluted in 3.2 μL of 0.1 M DTT (Euromedex) in a low-binding Eppendorf tube, and incubated on ice for 30 min. The actin mix is then centrifuged for 30 min at 20000 g at room temperature, and

supernatant is transferred in a new low-binding tube. Actin supernatant is then diluted in 3.2 μL of polymerization buffer (KCl (150 mM), MgCl2 (6 mM), Imidazole pH 7.4 (75 mM, Bio Basic Canada), MgATP (0.3 mM), to reach an actin concentration of 31 μM. This amount of pre-polymerized actin allows the production of five SLBs with the highest actin density; the protocol can be scaled up for bigger SLBs productions. Actin is then incubated 45 min at room temperature, and diluted with dilution buffer (KCl (50 mM), MgCl2 (2 mM), Imidazole pH 7.4 (25 mM, Bio Basic Canada)), to reach the expected final concentrations (4.40 μM, 2.20 μM or 0.44 μM). SLBs (in 100 μL TRIS HCl buffer) containing various molar proportions of biotinylated lipids are incubated by adding 10 μL of Streptavidin (MSD Millipore) at 0.1 mg/mL for 10 min and washed twice with 50 μL TRIS HCl Buffer. The streptavidin-bound SLBs (in 110 μL buffer) are then incubated with an addition of 10 μL of Phalloidin-XX-Biotin (Santa Cruz Biotechnology) at 1 μM for 10 min. Phalloidin-bound SLBs are washed twice with 50 μL TRIS HCl Buffer, and incubated with an addition of 10 μL of the previously described polymerizing actin for 20 min. Then 20 μL of Phalloidin Alexa Fluor 647 or 488 (Invitrogen) at 165 nM, for SLBs, is added to the solution (in the 150 μL buffer over the SLBs). SLBs are then incubated with actin overnight at 4 °C before measurements. Change in lipid mobility in the SLB after 18 h at 4 °C was controlled by single spot FCS on different SLB labeled either with 0.02% Atto-647N-PI(4,5)P$_2$ or Cy5.5-DSPE (from Avanti Polar Lipids, Inc) fluorescent lipid analogs (Supplementary Fig. 10, left graph). Similarly, we controlled that the addition of streptavidin and Phalloidin biotin without actin to the SLB, did not change the average lipid mobility (Supplementary Fig. 10, left graph) or Gag clustering at the surface of the SLB (Supplementary Fig. 10, right graph).

## Gag labeling and quantification
100 μL of HIV-1 myr(-)Pr55$^{Gag}$ protein (produced by J. Mak, Australia, as in[27]) is measured with NanoPhotometer® (Implen) at a 1.803 mg/mL (33 μM) initial concentration and incubated overnight at 4 °C under agitation with 1 μL Alexa Fluor 488 C5-maleimide (Invitrogen) at a 20 mM concentration in DMSO. This solution is transferred in Slide-A-Lyzer MINI Dialysis Device, 0.5 mL (Thermo Scientific) and incubated for 6 h at 4 °C under agitation in 15 mL Buffer; Tris (50 mM), NaCl (1 M), pH 8.0. The Buffer is changed, labeled myr(-)Gag is incubated again overnight at 4 °C, collected and stored at -20 °C.

## Jurkat cell sample preparation for FCS mesurement
Twenty-four hours post drug treatment, Jurkat T cells were washed and incubated with 30 nM of Atto647N-PI(4,5)P$_2$ for 5 minutes as in ref. 14, followed by excessive washes, for Atto647N-PI(4,5)P$_2$ diffusion measurements. Cells were then seeded on 1.5 H 25 mm slide coated with polylysine for 30 minutes just before aquisition.

## Spot variation fluorescence correlation spectroscopy (svFCS)
svFCS Experiments were performed on a Zeiss LSM780 confocal microscope (Zeiss, Iena, Germany) using an immersive 40X water objective equipped with a variable pupil coverage system to obtain different excitation laser waists. Argon 488-nm laser line was used for excitation of Rhodamine or Gag/Phalloidin Alexa Fluor 488, and HeNe 633-nm laser line for Atto647N-PI(4,5)P$_2$. Acquisition was controlled by the Zeiss Zen software. The laser waists for the different pupil coverage values were calibrated with the diffusion time of rhodamine (for 488 nm excitation) or Cy5 (for 633 nm excitation) in solution (D = 360 μm².s⁻¹ at 20 °C for rhodamine and 320 μm².s⁻¹ at 20 °C for Cy5 (based on Picoquant application note[69] and references therein)).

For each waist, at least 50 measurements of 10 seconds are made. Autocorrelation functions are analyzed using PyCorrFit 1.1.7 software[70] to extract the average decorrelation half time ($\tau_{1/2}$) for each probed waist. The svFCS diffusion laws were then established by plotting $\tau_{1/2}$ as a function of the square of the probed waists ($w^2$).

These diffusion laws were then fitted either a linear model $\tau_{1/2} = w^2/4D + t_0$ in the case of lipid membrane diffusion (Atto647N PI(4,5)P$_2$ in Jurkat T-cells and SLBs), or using equation 6 and 7 in ref. 43 in the case of the membrane partitioning HIV-1 Gag protein dynamics measurements to determine its membrane bound diffusion coefficient (D$_{bound}$) and the partition coefficient (Kp) in the different SLBs, using a MATLAB function and scripts previously described in ref. 43 (https://gitlab.inria.fr/hberry/gag_svfcs).

### Virus production

2.5 million of Human embryonic kidney cells (293THEK cell) were seeded in 10 ml of growth media 1 day before transfection. At 50-70% confluence, cells were transfected with calcium phosphate precipitate method, with 8 μg total quantity of plasmid for both pHIV-1 NL4.3 WT or pVSV-G + pHIV-1(i)GFPΔEnv (ratio 1:4). Culture media was changed 12 h post-transfection and cell culture supernatants containing viral particles (HIV-1 NL4.3 or VSV-G pseudotyped HIV-1(i)GFPdeltaEnv) were collected 48 hours post transfection. Supernatant was filtered through 0.45 μm and then purified by ultracentrifugation on a 25% sucrose cushion in TNE buffer (10 mM Tris-HCl [pH 7.4], 100 mM NaCl, 1 mM EDTA) at 100000 g, for 1 h 30 minutes at 4 °C, with a SW32Ti Beckman Coulter rotor. Dry pellet was resuspended with RPMI without serum, at 4 °C overnight. Viral titer was quantified by anti-HIV-1 p24 alphaLISA immunoassay (Perkin Elmer). Recombinant HIV-1 p24 protein is used for titer standard range.

### Infection and drug treatment

One million of activated PBLs or Jurkat T cell cultured in 2 ml RPMI media per well of 6 well plate or 0.1 million of cell in 200 μl RPMI per well of 96 well plate, were infected with 500 ng/ml HIV-1 p24, for 2 hours at 37 °C. Excess wash with PBS was performed to remove the unattached virion. 24 hours post infection, cells were washed and new media was added supplemented with drug. Infected cells were treated with drug for 24 hours. At 48 hours post infection, supernatant was collected. Viral release was quantified in HIV-1 p24 ng/ml by alphaLISA directly on supernatant collected from 96 wells plate. In the case of 6 wells plate infection, collected supernatant was clarified at 1400 g for first 5 minutes then 4500 g for other 5 minutes. Supernatant was loaded on a 25% sucrose cushion in TNE buffer and ultra-centrifuged at 100000 g, for 1 hour 30 minutes at 4 °C, with a SW55Ti Beckman Coulter rotor. The pellet was resuspended with TNE buffer 1X overnight at 4 °C. Viral release was estimated by performing anti-CAp24 immunoblot on cell extract as well as purified virions. Full (uncropped/unprocessed scans) western blots with molecular weight (MW) markers are provided in the "Western blots" Source Data file. Quantification of Gag pr55 and p24, as well as GAPDH are done using Fiji software. The percentage of viral release was calculated based on the following formula: % of viral release = Gag released/(Gag released + Gag intracellular normalized to GAPDH) *100 % of Gag maturation is = Capsid/(Gag + Capsid)*100 released in virion. Cell viability is measured by CellTiter 96 AQ (Promega ref G3581) kit when in vitro infection is designed in 96 well plates, or by trypan blue in the case of 6 well plate culture.

### siRNA electroporation

Using Amaxa 4D nucleofector machine and cell line electroporation Kit (Amaxa, Cat No V4xC1024), 1 million of Jurkat T cells were electroporated with 360 pmol of siRNA Arpin (same amount is used for the siRNA control). Electroporated Jurkat were then cultivated with 2 ml per well of RPMI without antibiotics in a 6-well plate.

### Flow Cytometry

F-actin and intracellular Gag-GFP measurements were performed by flow cytometry. Briefly, infected T cells expressing Gag-GFP were fixed with 4% PFA in PBS, and stained with phalloidin Alexa Fluor 647 or F-actin antibody with secondary conjugated to Alexa Fluor 647. After staining, samples are washed and resuspended with PBS. 20,000 events were analyzed by Novocyte/FlowJo.

### Immunoprecipitation assay

2.5 million HEK293T cells per 10 cm dish were transfected with 8 μg of pGag (Myc). pcDNA3.1 'empty plasmid DNA) is used as control. The cell medium was replaced 6 hours post-transfection. After 24 hours post-transfection, the cells were washed with phosphate buffer solution (PBS) and collected with Triton lysis buffer (50 mM TRIS-HCl [pH = 7.4]; 150 mM NaCl; 1 mM EDTA; 1 mM CaCl2; 1 mM MgCl2; 1% Triton, 0.5% sodium deoxycholate; protease inhibitor cocktail [Roche] one tablet/10 mL). The cells suspension was incubated on ice for 30 min and then centrifuged at 20000 g/15 min/4˚C. The supernatant was collected and total protein measurement assay was performed using Bovine Serum Albumin (BSA) (Thermo 23209) as standard range. For each condition, 1000 μg of total protein was incubated with or without anti-Arpin on a tube rotator overnight at 4˚C. Equivalent quantity of rabbit serum was used as control. 25 μL of beads (Dynabeads Protein A, Life Technologies) was added to each condition and incubated for 2 hours on the tube rotator at 4˚C. The samples were then washed five times with the lysis buffer, followed by addition of 20 μL 4xLaemmli's buffer to the beads. The samples were denatured at 95˚C for 10 min and then processed for Western blot.

### Sample preparation for confocal and F-actin content analysis

48 hours post infection, Jurkat T cell were washed one time with warm PBS and seeded on poly-l-lysine (Sigma) coated 12 mm round coverslips in microscopy phenol-red free medium L15 supplemented with 20 mM Hepes for 30 minutes at 37 °C. Cells were then fixed using 4% PFA + 4% sucrose in PBS for 15 min at room temperature, and quenched after in 50 mM NH$_4$Cl for 5 min. Samples were incubated with 1:20 dilution of phalloidin 647 Alexa Fluor (Invitrogen) overnight at 4 degree for actin labeling, and then washed with PBS. Prolong Gold antifade reagent (Invitrogen) is used to mount the slide 2 days before imaging. Z stack acquisitions manually selected from bottom to upper side of the cell (around 10 to 20 slices) are performed on Confocal LSM980 microscopy (MRI platform, CNRS Montpellier, France), with same laser power. F-actin quantification is performed using Fiji software after z-projection and F-actin intensity per cell using manual countering for each cell.

### Sample preparation for TIRF and STORM microscopy

48 hours post infection, Jurkat T cell were washed one time with warm PBS and seeded on poly-l-lysine (Sigma) coated 25 mm round #1.5 coverslips (VWR) in microscopy phenol-red free medium L15 supplemented with 20 mM Hepes for 30 minutes at 37 °C. Cells were then fixed using 4% PFA + 4% sucrose in PBS for 15 min at room temperature and quenched after in 50 mM NH$_4$Cl for 5 min. Samples were then washed in PBS. For STORM imaging, an immunofluorescence protocol was applied: Samples were incubated with Triton 0.2% in PBS for 5 minutes at room temperature, and washed then blocked for 15 min at room temperature using 2% BSA in PBS. Samples were stained using a 1:500 dilution of nanobody GFP booster for 1 hour at room temperature. Samples were washed three times for 5 min with PBS and stored in light light-protected container at +4 °C until imaged. For actin labeling, TIRF and STORM samples were incubated with 1:20 dilution of phalloidin 647 Alexa Fluor (Invitrogen) overnight at 4 degrees. Phalloidin was washed times with PBS before imaging. For Arpin labeling, samples were incubated first with 1:50 dilution of Anti-Arpin (Sigma ABT251) for 1 hour at room temperature, then washed three times with 2%BSA. This was followed by an incubation with 1:2000 dilution of secondary anti-rabbit antibody ATTO647N (Sigma) for 1 hour at room temperature. To correct the drift in the case of STORM, TetraSpeck microspheres 0.1 μm beads (Ref: T7279, Life

Technologies Corporation) were added to the samples for 5 minutes and then washed. STORM samples were mounted with the STORM buffer (Everspark buffer, Idylle Paris France) and then sealed with a sealing kit from Idylle. GFP nanobodies labeled Gag(i)GFP in the context of HIV-1 infectious CD4 T cells were suitable for STORM acquisition allowing us a precision of localization of 22 nm for Gag and 26 nm for F-actin (see Supplementary Fig.6A, 6B, 6C).

### TIRF and STORM Imaging
TIRF and Single-molecule localization microscopy was performed on a Nikon inverted microscope with an oil immersion objective 100×. STORM imaging was performed with 120 mW of 561 nm laser and 200 mW of 641 nm laser. Illumination was performed in TIRF-mode. For STORM aquisitions, 30,000 frames were acquired for each cell with 20 ms exposure time with 561 nm laser and 50,000 frames with 30 ms exposure time with 641 nm laser (adapted from[27]). Tetraspeck 100 nm beads (Life Technologies) were used as fiducial markers to correct for drift and chromatic aberration.

### TIRF and STORM analysis for virus assembly density quantification
TIRF acquisitions were segmented using Ilastik software 1.4.0. Intensity-based threshold was applied to identify clusters size and intensity. Fiji (ImageJ v1.53t) software was then used to identify particles size and intensity. SMLM acquisitions were analyzed using the ThunderSTORM plugin in Fiji (ImageJ v1.53t). Post-processing imaging was applied to first eliminate the background noise with a threshold within 50 nm radius. Next, duplicate localization were removed and repeating molecules within 20 nm were merge together. Drift was corrected by the drift correction module using fiducial markers applied on TetraSpeck beads. STORM localizations, found after post processing in ThunderSTORM reconstruction, were used to quantify viral clusters size using DBSCAN[71]. Intensity based threshold was applied after on each ROI to quantify the number of particles per surface using Fiji software (ImageJ v1.53t).

### TIRF F-actin and Gag cluster analysis
TIRF acquisitions were analyzed using Fiji (ImageJ v1.53t) software. 4 pixels x 4 pixels of Gag+ or Gag- area zone was selected to quantify local F-actin intensity. F-actin intensity per area was normalized to total F-actin intensity per cell in order to compare within cells and conditions.

### Dual color STORM analysis for F-actin and gag coordinates based colocalization (CBC) quantification
To assess the average colocalization of actin filaments and Gag assembly clusters, Gag images were first segmented by keeping only Gag localization belonging to the Gag clusters identified by DBSCAN. The CBC was then performed on these segmented Gag images taking 30 successive steps of 10 nm (300 nm) as the maximum searching distance (Rmax) to retrieve actin localizations from the localization belonging to Gag clusters (as in our previous study[27])

### Membrane flotation assay
The protocol was adapted from[16] and[39]. For each condition, 20 million Jurkat T cells were infected with VSV-G pseudotyped pHIV-1(i)GFPΔEnv virus particles and cells were collected 48 hours later. Cells were washed once with PBS and resuspend with Tris-HCl containing 4 mM EDTA and 1× complete protease inhibitor cocktail (Roche). Dounce homogenizer is used to fractionate the cell membranes and then 3 min 600 g centrifugation was used to obtain Post-Nuclear Supernatants (PNS). PNS, adjusted to 150 mM NaCl, was mixed first with 65% (wt/vol) of sucrose in TNE buffer in an ultracentrifugation Beckman tube. Then sucrose gradient was performed on top with 2.3 ml of 50% and 0.9 ml of 10% sucrose. An overnight ultracentrifugation was performed at

130000 g at 4 °C to allow membrane to flot with a SW55tiBeckman rotor. From top to the bottom, we collected 8 fractions of 500 µl. Western blot was then performed for analyzing viral and cellular protein content.

### Dual color STORM image analysis for visualization of virus clusters in actin meshwork
STORM images after reconstruction and post treatment on Thunderstorm (as described above) are used for this analysis. Actin image as well as filtered clusters images are analysis using Fiji software. Watershed segmentation function followed by skeletonize plugin were applied to obtain actin skeleton representing actin meshwork.

### Statistical tests
Data were analyzed using Prism software (GraphPad) and MATLAB. Mann-Whitney tests were applied to compare the different values. In the main text, unless specified, data are presented as mean value +/-SEM

### Reporting summary
Further information on research design is available in the Nature Portfolio Reporting Summary linked to this article.

## Data availability
All relevant data supporting the key findings of this study are available within the article and in Supplementary Information files. Source data are provided as a Source Data file in a ZIP folder containing the data of each graph and in Supplementary information for the uncropped Western blots. Due to size constrains, relevant raw data of FCS and single molecule localization microscopy (STORM) are provided and available at the following address https://doi.org/10.5281/zenodo.8366334, the remaining raw data are available upon reasonable request. Source data are provided with this paper.

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

## Acknowledgements

The following reagent was obtained through the NIH HIV Reagent Program, Division of AIDS, NIAID, NIH: Human Immunodeficiency Virus Type 1 (HIV-1) NL4-3 Gag-iGFP ΔEnv Non-Infectious Molecular Clone, ARP-12455, contributed by Dr. Benjamin Chen. We thank Christophe Leterrier (Marseille, France) for the first introduction to F-actin imaging by STORM many years ago and H. Berry (Lyon, France) for the Matlab code of FCS diffusion law analysis. RD is a recipient of a SIDACTION PhD fellowship. We thank CEMIPAI facility for STORM/TIRF microscopy in BSL3, and MRI facility for FCS and confocal microscopies. The study was supported by CNRS, SIDACTION #LS200599-DM, ANRS #248756-DM (French Agency for AIDS, Hepatitis Research) granted to DM. CF and DM are members of the CNRS Imabio consortium.

## Author contributions

R.D. performed cell culture, infection, sample preparation for biochemistry and microscopy, siRNA, IP, acquisition and quantification of confocal, TIRF-M and STORM microscopies; J.M. produced and purified recombinant HIV-1 Gag protein; E.B. and C.F. performed in vitro model membranes experiments and sv-FCS analysis in vitro and on T cells; R.D. and C.F. performed STORM imaging data analysis and quantification; D.M. directed the study. D.M., R.D. wrote the original draft manuscript; R.D., C.F., and D.M. edited figures and manuscripts. DM raised funding.

## Competing interests

The authors declare no competing interests.
