## [Peer Review File · Nature Communications]

HIV-1 diverts cortical actin for particle assembly and releaseREVIEWER COMMENTS

Reviewer #1 (Remarks to the Author):

The authors of this study investigate the role of cortical actin in HIV-1 assembly, specifically focusing on the possible involvement of branched actin. The data suggest that actin debranching increases HIV-1 release and the number of individual HIV-1 assembly clusters present at the plasma membrane. Some in vitro support is provided. Finally, the authors suggest that Arpin, an actin debranching factor and Arp2/3 inhibitor, interacts with Gag.

While the authors present some interesting data, in general most of the effects reported here are quite small, in many cases barely rising to the level of statistical significance. Additional comments are provided below.

1. Many of the experiments were performed with Gag containing an internal GFP insertion (GagiGFP). This modified Gag is well established to be defective for membrane binding and virus assembly; as a result, most groups combine the GFP-labeled Gag with unlabeled Gag to mitigate this defect. This is not done here, causing some concern about the validity of some of the results, in particular the membrane flotation data which show a rather low percentage of Gag in association with membrane.
2. Fig. 1. An effect of latrunculin on HIV-1 particle production in the Jurkat T-cell line is only apparent at the lowest concentration used (0.25 μ M). Similarly, the effect in primary T cells is greater at lower than at higher concentrations. This phenomenon is not explained.
3. Fig. 2. The reported increases in particle release observed upon treatment of virus-producing cells with the actin debranching inhibitor CK666 are quite small (e.g., 20% in some cases). The legend states $N=4$, but it is not clear if this means 4 independent experiments or technical replicates.
4. Line 127/128 states "Actin debranching drug CK666 increases HIV-1 release from infected CD4+ T lymphocytes without impairing virus maturation". However, no data on virus maturation are shown.
5. Fig. 6F. The authors show 4% binding of arpin to cell membranes in uninfected cells and 7% in infected cells ($p=0.0432$). I'm not convinced that this small effect is real given the very low % membrane binding and the low quality of the blots.
6. Fig. 6G. The reported co-IP of Gag and arpin is not convincing. The same upper band (of approximately Gag MW) is also seen in lane 5, which uses control rabbit serum.

Reviewer #2 (Remarks to the Author):

The work by Dibsby et al. titled 'HIV-1 diverts actin debranching mechanisms for particle assembly and release in CD4 T lymphocytes' reports on the role of actin cortex organisation and density for the assembly and release of HIV-1 Gag-GFP positive particles using a combination of live cell and in-vitro assays.

The authors find that using a range of drugs reducing the assembly and stability of actin filament networks in cells along the CDC42/ Rac1 – Arp2/3 activation pathway, such as Cdc42 inhibitor III, CK666, Lat B, Jasplakinolide, the number of Gag1-GFP particles at the plasma membrane of T-cells increases and that the release of Gag1-GFP particles is increased compared to untreated control cells. Close inspection of actin cortex organization and Gag1-GFP particle distribution by TIRF- and STORM microscopy seems to suggest a reduction of actin network density at sites of Gag1-GFP particles. This leads to the idea that actin network clearance or debranching of the Arp2/3 actin cortex is required to allow Gag1-GFP particle assembly and release. The authors employ a minimal reconstituted system of actin filaments linked to PIP2 containing supported lipid bilayers using biotin-phalloidin, streptavidin and biotinylated lipids to observe the binding of Gag1 to the lipid bilayer and formation of Gag1-clusters. Indeed, they observe a reduced Gag1-cluster formation in high-density actin networks coinciding with a reduced PIP2 diffusion. Last, the authors identify the actin debranching factor Arpin as important for Gag1-GFP particle release in T-cells.

The work puts forward an interesting link between HIV-Gag1 particle release and actin network

organisation using a variety of experimental approaches. Though the core of the data and story is very convincing, some of the experimental procedures would require clarification and additional controls.

Major points of concern:

- It is nice to see a work that employs live cell experiments and reconstituted minimal systems to test ideas and obtain further insights into possible mechanisms of Gag1-particle assembly. However, the procedures described for the preparation of the SLBs seem at least very unconventional. The authors combine extrusion with sonication to make SUVs, which is as such ok, but the sonication time of 16 min seems excessive and can cause heating up of the lipid mix. Especially with the short lived PIP2, such treatment seems unfavourable. Unless I missed such experiments mentioned in the manuscripts, control experiments of GAG1 binding on SLBs without PIP2 would be helpful to ensure the importance of PIP2.
- Then it is written, that SLBs with actin and Gag1 are incubated over night at 4C before imaging. This incubation time is unnecessary long and it is highly doubtful, how much one can trust the SLBs and the functionality of PIP2 after such long time. Cooling down and warming up SLBs is reported to increased defects in SLBs and in increased tubulation. This is particularly true for phase separation lipid compositions as used by the authors. The authors should show control experiments of SLB motility using FRAP / FCS on another labelled lipid, such as Rhodamine-PE, to demonstrate overall mobility of bulk lipids. It would also be important to do control experiments of Gag1 clustering and mobility in SLBs that contain phalloidin-biotin, streptavidin and biotin lipids but no actin, to rule out effects of biotin-lipid cross-linking by streptavidin.
- based on the in vitro work, the authors make the argument that actin debranching is required to allow higher PIP2 mobility in cells allowing assembly of Gag1 particles. However, the data presented here remains correlative as the mechanisms of actin binding to the SLB via phalloidin-biotin is very different from the physiological situation. It would be crucial that the authors measure the mobility of PIP2 or Gag1-GFP in cells with and without CK666 treatment to test their hypothesis.
- The quantification of actin with Phalloidin staining after treatment with Jasplakinolide will not work well because Phalloidin and Jasplakinolide bind the same site on actin.

Minor points:

- The volumes used for the preparation of the SLB samples with actin and Gag1 (50-10 ul) seem very low for a 7 mm diameter well. And it would be good to clarify in the actin polymerisation protocol, what the final volume of the buffer is in each well. E.g. when adding 20 ul Phalloidin, is this only the 20 ul or are the 20 ul added to some amount of buffer already present ?
- figures have to be checked carefully. Fig 4C/D should show the region of the zoom. Fig 5 G is missing units at the axis label, Fig 5 E/F should use another colour code than Red/Green as this is not suitable for colour blind people.
- the writing needs in some regions improvement. In the beginning of results, the authors should name the Rac1 and CDC42 inhibitors used, when writing of IC50 values, they should say 150 uM of what. Page 5, 'Quantification of HIV-1 p24 release showed a 50% significant increase ...' doesn't make much sense. Page 6, doubling time gets faster or shorter, not lower. And many more such instances.
- Page 7: '... f-actin mean intensity of gag+ area of value 0.9435...' of what? Given the std of 0.16, there are too many digits given. The precision doesn't allow to write all four decimals. It should be 0.94 +/- 0.16 and then a unit should be written.

Reviewer #3 (Remarks to the Author):

Dibsy and coworkers describe studies that represent a logical and appropriate follow-up to their 2015 J. Virology paper (PMID: 26018170). In the previous study, the group reported that activated Rac1 and the IRSp53-Wave2-Arp2/3 signaling pathway were involved in HIV-1 Gag membrane localization and particle release in T-cells as well as a Gag-dependent role for actin debranching and polymerization. In this current report, the authors have assessed if Arp2/3 mediated branched actin is involved in HIV assembly at the membrane of infected T-cells. They

show that actin debranching increases HIV particle release as well as the number of HIV assembly clusters at the membrane. They further show that Gag assembly is found in areas with less F-actin, which they further demonstrated with in vitro studies. Arpin (a Arp2/3/ inhibitor) was recruited by Gag and promoted virus assembly.

Major comments:

1. The general topic understudy in this manuscript is both timely and important. Here, the authors provide a logical follow up to their previous work and have uncovered additional details to help better explain how the IRSp53-Wave2-Arp2/3 signaling pathway is involved in HIV assembly. The observation of virus release being associated with actin debranching is important. As outlined below, there are number perceived weaknesses that limit the interpretative power of the dataset.
2. The use of drugs that affect actin dynamics (ie, Fig 2 and Jasplakinolide, LatrunculinB), while are being investigated at concentrations that appear to minimize cell toxicity), would be enhanced if they were complemented with other experimental approaches as the drugs could have off-target effects that leading to the observations being made.
3. The latrunculin data seems somewhat confusing. If decreasing F-actin lowers particle production, it would seem to go against the idea that increasing actin debranching helps assembly. They seem to have a more complex model of the role of "actin treadmilling". This data and the applicability of the complex treadmilling modeling needs to be better clarified.
4. The experiments with CK666 (Fig 3 & 4), while informative, raise a similar concern that off-target effects from the drugs could be influencing the results being reported.
5. Fig 5 uses an in vitro model to investigate Gag clustering on model membranes. The strength of this model is that they can manipulate conditions to access the effects on labeled Gag clustering. It is not clear whether actin formation on the model membranes is relevant to what is seen along the plasma membrane of cells. The general data shown tends to correlate with actin influencing Gag clustering, but it is unclear how well this in vitro model simulates sites of Gag clustering in cells regarding form of actin and membrane lipid composition.
6. The authors targeted the expression of Arpin (which is a actin debranching factor) in Fig 6. Reducing Arpin expression by 40% resulted in a very modest reduction in HIV release. The authors state that this results in a reduction of 30% in HIV release, while the actual histogram in Fig 6C looks no better than 20%, and with the variability of the assay this difference may be less than this. Overall, the trend supports the conclusion that the authors are seeking to draw. However, the data is not strongly supportive of the conclusion.
7. It is unclear if the Gag clustering observed by the authors can be differentiated from released particles immobilized under the cell in their microscopy and in their TIFR images. It would be expected that released particles would swamp out any population of Gag clustering at the membrane. The rate at which Gag clustering occurs is quite fast, and the odds of detecting Gag clustering prior to particle release would be challenging.

Minor comments

1. The fixation protocol for the microscopy imaging is a bit concerning. 4% PFA can result in a significant loss of soluble proteins, like Gag and G-actin.
2. Cytoplasmic HIV Gag diffuses slowly in part due to the presence of the cytoskeleton. Disrupting the cytoskeleton might lead to a faster loss of soluble Gag from cells, then normalizing intensity as described by the others in several instances could lead to the observation of relatively more Gag at the membrane. The diffusion timescale for Gag to be released from a damaged cell is probably several minutes or more, which is about the time used in fixation.
3. More details are needed for the model for how diffusion of HIV Gag on membranes can influence the initiation of Gag clustering. The dynamics of this process would appear to be fairly complex, and it seems to be appropriate to incorporate a reaction diffusion model.

Manuscript Number: **NCOMMS-22-50340A**

Title « **HIV-1 diverts actin debranching mechanisms for particle assembly and release in CD4 T lymphocytes.** »

Rayane DIBSY et al.

Responses to the Reviewers' comments (in blue color):

Reviewer #1 (Remarks to the Author):

The authors of this study investigate the role of cortical actin in HIV-1 assembly, specifically focusing on the possible involvement of branched actin. The data suggest that actin debranching increases HIV-1 release and the number of individual HIV-1 assembly clusters present at the plasma membrane. Some in vitro support is provided. Finally, the authors suggest that Arpin, an actin debranching factor and Arp2/3 inhibitor, interacts with Gag.

While the authors present some interesting data, in general most of the effects reported here are quite small, in many cases barely rising to the level of statistical significance. Additional comments are provided below.

We would like to thank the reviewer for the comments. In order to convince the reviewer 1, we have performed new set of experiments to increase their total number, enhancing statistical significance. These complementary experiments are now provided in the new manuscript. We hope they will convince him/her that our observations (at the nanoscale level) are (statistically) significant.

In addition, and according to Nature Communication policy, we now provide all the comparison based on statistical tests which are detailed in the Material and Methods section. The associated p-value ranges are noted in the figure legends while the exact p-values are specified in the text when important.

1. Many of the experiments were performed with Gag containing an internal GFP insertion (GagiGFP). This modified Gag is well established to be defective for membrane binding and virus assembly; as a result, most groups combine the GFP-labeled Gag with unlabeled Gag to mitigate this defect. This is not done here, causing some concern about the validity of some of the results, in particular the membrane flotation data which show a rather low percentage of Gag in association with membrane.

It is true that in the previous literature, it was said that Gag(i)GFP assembly was incomplete. However, since 2007, the pNL43-Gag(i)GFP was regularly improved creating a molecular clone of HIV-1 (NL43) with an insertion cleavage surrounding the GFP that was totally compatible with HIV-1 Gag assembly with a correct binding of Gag to the cell plasma membrane, and correct particle assembly and release was reported, as well as viral synapse engagement (see references 1, 2, 3 and 4 below). However, to convince the reviewer, a Supplemental Figure 1 was added to the manuscript, describing our production of HIV-1(i)GFP infectious particles as compared to WT HIV-1 (pNL43) and their level of maturation. We also validate HIV-1(i)GFP infection in our model Jurkat T cells (Supplemental Figure 1 F, 1G). Our results show that HIV-1(i)GFP particles are produced with a good yield, they are matured and infectious as the WT ones are. This is also now described in the result section – page 4 - of the revised manuscript.

1. Hubner, W. et al. Sequence of human immunodeficiency virus type 1 (HIV-1) Gag

localization and oligomerization monitored with live confocal imaging of a replication-competent, fluorescently tagged HIV-1. *J Virol* 81, 12596-12607 (2007).

2. Chen, P., Hubner, W., Spinelli, M.A. & Chen, B.K. Predominant mode of human immunodeficiency virus transfer between T cells is mediated by sustained Env-dependent neutralization-resistant virological synapses. *J Virol* 81, 12582-12595 (2007).

3. Hubner, W. et al. Quantitative 3D video microscopy of HIV transfer across T cell virological synapses. *Science* 323, 1743-1747 (2009)

4. Favard, C. et al. HIV-1 Gag specifically restricts PI(4,5)P2 and cholesterol mobility in living cells creating a nanodomain platform for virus assembly. *Science Adv.* eaaw8651 (2019)

We have previously addressed this issue in Jurkat T lymphocytes for HIV-1 Gag transfected or HIV-1(i)GFP infected cells and showed that the HIV-1(i)Gag-GFP was showing assembly clusters at the T cell plasma membrane when only expressing Gag(i)GFP (see section 6 in Supplementary Information in Favard et al., *Sci Adv* 2019 eaaw8651).

The membrane flotation technique applied to HIV-1 infected or Gag-transfected in Jurkat T cells always show a very low percentage of Gag membrane bound fraction, even without (i)GFP, (see Figure 1 in Thomas et al., *JVI* 2015, doi: 10.1128/JVI.00469-15) : we think that it is not due to Gag membrane binding ability in Jurkat T cells (as these cells produce high number of viral particles – see Supplemental Figure 1 of this manuscript) but rather due to the cell type (Jurkat T cells versus transfected 293THEK cells (see Inamdar et al., *eLife* 2021)).

2. Fig. 1. An effect of latrunculin on HIV-1 particle production in the Jurkat T-cell line is only apparent at the lowest concentration used (0.25 μ M). Similarly, the effect in primary T cells is greater at lower than at higher concentrations. This phenomenon is not explained.

We thank the reviewer for this remark. We realized that this can make the interpretation somehow confusing, therefore, in order to get more insight into it, new experiments were performed. The lower concentration (0.25 μ M of Latrunculin B - LatB) was further studied and compared to the higher concentration (1 μ M of LatB) as reported now in the new Figure 1. Thanks to the increased number of experiments, as well as additional new type of experiments, the results show LatB treatment, that fosters F-actin depolymerization, results in enhancing HIV-1 release (as shown by different concentration of LatB, immunoblots or alphaLisa) in infected Jurkat T cells (new Figure 1) and in primary T cells (new Supplemental Figure 1B). These additional results are now further explained in the first result section (pages 4-5).

3. Fig. 2. The reported increases in particle release observed upon treatment of virus-producing cells with the actin debranching inhibitor CK666 are quite small (e.g., 20% in some cases). The legend states N=4, but it is not clear if this means 4 independent experiments or technical replicates.

As this is an important point, not only we clearly stated what was independent experiments and number of replicates (technical or not) but also, we repeated experiments and therefore added new replicates. The numbers of experiments (N) and replicates (n) were added in the main text (result section) and in all the figure legends.

We now have $3 < N < 10$ independent experiments and $6 < n < 15$ replicates for the different conditions tested here. As expected, this strongly improve our confidence in the effect, although difference could look quite small (20% difference in absolute value (from 50 to 70% of particle release)) in the worst case. Indeed, even in the WT HIV-1 conditions infected primary blood lymphocytes, when we

test the percentage of viral release, we now have a p-value=0.0164 (Two-tailed Mann-Whitney test) indicating a significant difference. These p-values drops down to 0.002 to 0.004 in the two other conditions testing the effect of CK-666 on viral release.

4. Line 127/128 states “Actin debranching drug CK666 increases HIV-1 release from infected CD4+ T lymphocytes without impairing virus maturation”. However, no data on virus maturation are shown.

We were certainly not very clear in the writing of Figure 2 results in the previous version of the manuscript. We have now clarified this statement in the result section 2 (page 8) and explain the calcul of particle maturation in Material and method section (page 20). The maturation of Gag in CK666-treated (or not) cells and in associated released viruses can be observed from the western blots depicted in Fig 2 F/G; From this blots, one can calculate that the ratio p24Capsid (matured Gag) over p55Gag (immature) is not altered by the CK666 treatment in infected Jurkat T cells or in primary T cells. However, to help the reader, and to consider the reviewer’s comment, we calculated this ratio and incorporated the values in Figure Supplemental 5. The results confirm that this ratio is not altered upon CK666 treatment as compared to DMSO treatment, thus CK666 is not impairing virus maturation.

5. Fig. 6F. The authors show 4% binding of arpin to cell membranes in uninfected cells and 7% in infected cells (p=0.0432). I’m not convinced that this small effect is real given the very low % membrane binding and the low quality of the blots.

We agree that this value could seem very low, but it is important to note that it doubles in the presence of the virus. Although low, this value is notable because Arpin is not supposed to be a membrane binding protein (as reported in Dang et al Nature 2013 PMID: 24132237 : « *Arpin is very less present at the cell membrane, it is seen only on the tip of lamellipodia* »). However, we performed several independent experiments (N=3 independent membrane flotation assays – see figure R1 below) and always found at least a doubling amount of Arpin at the cell membrane in infected T cells (2.3 fold increase, with a p-value=0.0432, two-tailed unpaired t-test).

Thus, in our opinion, our data undoubtedly show that upon HIV-1 infection, a portion of cellular Arpin is displaced towards the cell membrane. In addition, to reinforce our observation and further convinced the reviewer 1, we performed a new experiment using STORM/TIRF microscopy to image the presence of Arpin at the cell membrane of infected T-cells as compared to non-infected. New Figure 6H depicts STORM imaging revealing that the number of Arpin localization/μm² at the T cell membrane increase significantly (p-value <0.0001) upon HIV-1 infection. This result is reinforcing the membrane flotation assay results shown in Figure 6 F/G.

Figure R1: 3 raw immunoblots showing membrane binding ability of Arpin upon Gag expression in T cells. One can see Arpin membrane binding in infected T cells as compared to non-infected Jurkat T

cells. Fraction of Membranes are represented as : 1, 2 and 3, and Fractions of Cytosol are represented as : 6, 7 and 8. This results corresponds to Figure 6G.

6. Fig. 6G. The reported co-IP of Gag and arpin is not convincing. The same upper band (of approximately Gag MW) is also seen in lane 5, which uses control rabbit serum.

Thanks to the reviewer's comment, we have improved our co-IP between Gag and Arpin. We repeated the IP several time (see below – Figure R2) and we can undoubtedly affirm that there is a significant IP-band of Gag when IP with an anti-Arpin antibody that is neither present in the control rabbit serum, nor without antibody (sometimes a faint band appears but this is only due to IP background and is very different from the Gag band – compare lanes 1 and 3). Even with a high IgG concentration, we have a very significant difference between the band intensities (lane 1 versus lane 3). So, we are convinced that Gag co-IP with Arpin.

Figure R2: Immunoblot showing the immunoprecipitation of HIV-1 Gag with antibody anti-Arpin (at the left) and their correspondent Flowthrough at the right. 1: transfected cell lysate incubated with anti-Arpin. 2: transfected cell lysate incubated with no Arpin antibody. 3: transfected cell lysate incubated with rabbit serum.

Reviewer #2 (Remarks to the Author):

The work by Dibsby et al. titled 'HIV-1 diverts actin debranching mechanisms for particle assembly and release in CD4 T lymphocytes' reports on the role of actin cortex organisation and density for the assembly and release of HIV-1 Gag-GFP positive particles using a combination of live cell and in-vitro assays.

The authors find that using a range of drugs reducing the assembly and stability of actin filament networks in cells along the CDC42/ Rac1 – Arp2/3 activation pathway, such as Cdc42 inhibitor III, CK666, Lat B, Jasplakinolide, the number of Gag1-GFP particles at the plasma membrane of T-cells increases and that the release of Gag1-GFP particles is increased compared to untreated control cells. Close inspection of actin cortex organization and Gag1-GFP particle distribution by TIRF- and STORM microscopy seems to suggest a reduction of actin network density at sites of Gag1-GFP

particles. This leads to the idea that actin network clearance or debranching of the Arp2/3 actin cortex is required to allow Gag1-GFP particle assembly and release. The authors employ a minimal reconstituted system of actin filaments linked to PIP2 containing supported lipid bilayers using biotin-phalloidin, streptavidin and biotinylated lipids to observe the binding of Gag1 to the lipid bilayer and formation of Gag1-clusters. Indeed, they observe a reduced Gag1-cluster formation in high-density actin networks coinciding with a reduced PIP2 diffusion. Last, the authors identify the actin debranching factor Arpin as important for Gag1-GFP particle release in T-cells.

The work puts forward an interesting link between HIV-Gag1 particle release and actin network organization using a variety of experimental approaches. Though the core of the data and story is very convincing, some of the experimental procedures would require clarification and additional controls.

Major points of concern:

-It is nice to see a work that employs live cell experiments and reconstituted minimal systems to test ideas and obtain further insights into possible mechanisms of Gag1-particle assembly. However, the procedures described for the preparation of the SLBs seem at least very unconventional. The authors combine extrusion with sonication to make SUVs, which is as such ok, but the sonication time of 16 min seems excessive and can cause heating up of the lipid mix. Especially with the short lived PIP2, such treatment seems unfavourable. Unless I missed such experiments mentioned in the manuscripts, control experiments of GAG1 binding on SLBs without PIP2 would be helpful to ensure the importance of PIP2.

We thank the reviewer for his/her comment. Indeed, SUV preparation has been long time directly achieved from the MLV solution using sonication in bath or in ice with a tip sonicator.

Avanti polar recommend sonication to last for 5 to 10 min, but at the same time they provide a link to the Morissey (University of Illinois Champaign-Urbana) lab whom's protocol suggest sonication to last for 15 to 30 min. The literature is very controversial about the effect of sonication on lipids structure. Some studies have shown changes in temperature transition of DPPC

(doi.org/10.1016/j.bbrep.2020.100764), suggesting chemical modification of lipids, when submitted to tip sonication longer than 12 min. However, the tip sonication reference in the mat & met section indicates that the power of the sonicator is 500W to 700W where ours, a bath sonicator is only 200W, with a much larger volume to dissipate the increase in temperature. We failed to find a robust study taking into consideration power, time and volume of sonication to give a precise idea of the impact of sonication on lipid structure. However, we chose to perform 16 min sonication on top of extrusion, since we wanted to generate as most as possible a monodispersed high radius curvature SUVs solution to facilitate the fusion of our highly enriched negatively charged PS and PIP2 SUVs without using Ca²⁺ ion (that will screen HIV-1 Gag binding to the membrane). Nevertheless, we agree with the fact that 16 min sonication could seem excessive since we already performed extrusion, and might impact the PIP2 molecular percentage (hydrolysis?) left in the SUVs after such a treatment. Therefore, we used spot variation FCS and analyzed the related diffusion laws with our

recently published semi-empirical model (equations 5 & 6 in Mouttou et al., Biophys J, 2023, ref 42 of the main manuscript), to disentangle the binding partition from the membrane diffusion of the protein contribution in the fluorescence fluctuations. We compared the values of Gag binding to our SLB membrane prepared with the protocol use here (including overnight at 4°C, followed by re-equilibration at room temperature, (dark blue bars)) to the protocol we used for our Biophys. J. paper, in which we have prepared PIP2/PS containing membrane and only PC containing membranes (cyan bars) that followed the same protocol except the overnight incubation. Since PIP2 is the binding targeted lipid for Gag, for a constant amount of Gag injected in the solution, based on mass action law, the percentage of Gag bound to the membrane will mainly depend on the available PIP2 in the SLB. Finally, this binding can be compared to another binding assay, performed on only extruded MLV, where, Gag is bound to 200 nm LUVs made with the same lipid composition. From the graph below, it can be seen that the binding to the SLB is systematically estimated (with our FCS measurements) to be lower than the binding to the 200 nm LUV performed with a different assay. This could be due to the excessive sonication, but this could also be due to an underestimation of the binding by our FCS measurements. As it can be seen that even with unbinding lipids (EggPC) we estimate a lower binding partition. Most importantly, we observe that the Gag binding to the different PIP2 containing SLBs with increasing amount of actin, prepared with this protocol still are at least twice the one we obtained with EPC alone, suggesting that these SLBs still contain PIP2 on which Gag binds.

-Then it is written, that SLBs with actin and Gag1 are incubated over night at 4C before imaging. This incubation time is unnecessary long and it is highly doubtful, how much one can trust the SLBs and the functionality of PIP2 after such long time. Cooling down and warming up SLBs is reported to increased defects in SLBs and in increased tubulation. This is particularly true for phase separation lipid compositions as used by the authors. The authors should show control experiments of SLB motility using FRAP / FCS on another labelled lipid, such as Rhodamine-PE, to demonstrate overall mobility of bulk lipids. It would also be important to do control experiments of Gag1 clustering and mobility in SLBs that contain phalloidin-biotin, streptavidin and biotin lipids but no actin, to rule out effects of biotin-lipid cross-linking by streptavidin.

The overnight incubation at 4°C was implemented to achieve slow actin polymerization while preventing degradation/destabilization of the actin filaments at the same time. We hypothesis that, as 99.9% of the molecules are natural lipids (Brain PS, Brain PIP2, Egg PC), containing a large and mixed variety of saturated and (poly) unsaturated acyl chains of different lengths, the SLB will overall stay in liquid state, without experiencing an irreversible transition or macro phase separation and that, by exposing them to room temperature long time enough, the equilibrium, including possible microphase separations, when present, will be restored to their original values. Additionally, this lipid composition is not known to exhibit strong lateral phase separation (no cholesterol, no SPM, only natural lipids with at least 30% of charged lipids inserted into neutral polar heads).

Nevertheless, as suggested by the reviewer, we performed classical FCS of DSPE-Cy5.5 on our EPC/BPS/BPIP2 SLBs, before (fresh) and after 18 hours of incubation at 4°C that exceed the overnight. We did that for EPC/BPS/BPIP2 mixture and for the same composition with 1% biotinylated lipids+streptavidin and phalloidin biotin. In addition, we also did the same using Atto-647N PI(4,5)P2 as a lipid analogue. 5 to 10 different positions in the SLBs were tested to ensure for the absence of macro-phase separation, each recording from 20 to 40 correlograms. We could not detect a strong decrease in our lipid analogues mobility, suggesting that the overall mobility in SLBs of this lipid composition is not affected by a 18h stay at 4°C.

These results are now added as Supplementary Figure 10, and a sentence has been added in the Material and Methods section.

The Atto-647N PIP2 diffusion coefficients given in the original version of the paper and now depicted in new Fig. 5C were obtained in SLB prepared with this protocol and, according to their values, these lipid analogues already seemed fully mobile (Ld phase) even after one night at 4°C. This is also confirmed now, since the “no-actin” SLBs have an average diffusion coefficient in the range of the one we found in our additional experiments (before and after 18h) using single spot FCS on Atto-647N PIP2.

Finally, thanks to the diffusion law we performed, we could determine the intercept value at null waist for the different conditions. From these values, we can observe that there is no highly positive intercept value of Atto-647N PIP2 in each condition, suggesting that lipid lateral phase separation where the Atto647N-PIP2 might partition is not present in our SLBs. On the opposite, the negative intercept values of Atto-647N PIP2 diffusion laws in the presence of increasing actin concentrations, reflects its trapping in opened meshes by the (actin) fences as it has been shown previously (Wawreczynieck et al., ref 43 of our main manuscript, KI Lee, BMC Biophys., 2014 (doi.org/10.1186/s13628-014-0013-3)). These intercept values have now been added in new Fig 5D. Interestingly, the work from Wawreczynieck et al. also showed that the value of w^2 at $\tau=0$ (x-axis intercept of the diffusion law) gives the size of the meshwork which is impeding the diffusion. In the case of the 4.4 μM actin with 1% mol biotin SLBs, we found here $\langle w^2(t=0) \rangle = 0.017 \mu\text{m}^2$ (average of 3 different SLBs), which correspond (for a square mesh) to an average value of 130 nm side length for a mesh unit. This is line with the peak value of the mesh (120 nm) observed by Honingman et al. (ref 44 of our main manuscript), using the same molecular percentage of biotin anchors and imaging, with STED microscopy, the actin meshwork in fresh SLBs. This again strongly suggests that even after one night at 4°C the structural properties of our SLBs (mesh size, fluidity...) remains the same.

These different results strongly suggest that the SLB we made with this protocol in this study, despite possible too long exposure to sonication or possible phase separation (though totally unexpected with this composition) or generation of tubules (not seen under the microscope) are still functional for our diffusion, binding and self-assembly measurements on purified Gag protein.

The 4°C overnight incubation with Gag is an error in the Mat & Met. due to unclear phrasing in this sentence. Gag was indeed added to the solution only 30 minutes before FCS measurements, not during the whole night. This has now been corrected in the Mat & Met section.

Finally, since Honingman et al. (ref 44), have previously shown that diffusion of lipids was similar in fluid SLBs without actin, in the absence or in the presence of biotinylated lipid up to 1% mol, streptavidin and biotin phalloidin, and since we also observed the same in this study, we only checked that Gag clustering in the absence of actin, but in the presence of biotinylated lipids and streptavidin and phalloidin biotin was similar. We observed no difference in this condition compared to the no actin, no biotin, no streptavidin condition as depicted below (see aside). This is now added in Supplementary Figure 10.

-based on the in vitro work, the authors make the argument that actin debranching is required to allow higher PIP2 mobility in cells allowing assembly of Gag1 particles. However, the data presented

here remains correlative as the mechanisms of actin binding to the SLB via phalloidin-biotin is very different from the physiological situation. It would be crucial that the authors measure the mobility of PIP2 or Gag1-GFP in cells with and without CK666 treatment to test their hypothesis.

We deeply acknowledge the reviewer for this remark, since, as he/she pointed out, this is crucial for supporting the mechanism in the physiological situation and makes the paper even stronger. For this reason, we performed spot-variation FCS diffusion laws on Jurkat T-cells incubated with the lipid analogue Atto-647N PIP2, using the same protocol for lipid insertion that the one we used in (Favard et al., Sci Adv 2019, ref 14 of our main manuscript). The results are now added to Fig. 5C for the diffusion coefficient and Fig. 5D for the intercept value at null waist. Interestingly we observe clear and significant difference in PIP2 diffusion in the absence (DMSO) or presence of CK666. Moreover, the main parameters of PIP2 diffusions obtained with the svFCS diffusion laws, ie. D and t_0 behaves similarly in cells and SLB, strongly supporting the capacity of our SLBs to transpose the observed mechanism on Gag to physiologically relevant samples.

We performed sv-FCS diffusion laws of PIP2 instead of GFP-Gag, since the later had to be performed at very low cytosolic concentration ($10 < c < 50$ nM as we did in ref 42). We chose this for two distinct reasons. First, we showed in Mouttou et al. (42), that, in cells, the membrane bound diffusion coefficient is poorly estimated, unless additional labelling of the plasma membrane to find the correct z-position to acquire the fluorescence fluctuation, this is now stated in the main text of the manuscript in the result section (page 11). Second, because the Jurkat T cells used here exhibit strong natural fluorescence when excited at 488nm, from 500 to 600nm, this natural fluorescence contributes to the photon correlation and affects the decorrelation time measured when we use small concentrations of Gag-GFP to correctly determine K_p and D_{bound} .

-The quantification of actin with Phalloidin staining after treatment with Jasplakinolide will not work well because Phalloidin and Jasplakinolide bind the same site on actin.

We thank the reviewer for his/her comment. We made F-actin quantification with an anti-F-Actin antibody upon Jasplakinolide treatment using FACS that is now represented in Supplemental Fig.3A.

Minor points:

-The volumes used for the preparation of the SLB samples with actin and Gag1 (50-10 ul) seem very low for a 7 mm diameter well. And it would be good to clarify in the actin polymerisation protocol, what the final volume of the buffer is in each well. E.g. when adding 20 ul Phalloidin, is this only the 20 ul or are the 20 ul added to some amount of buffer already present?

We thank the reviewer for that comment; this was corrected and added in the Material and Method section accordingly. Indeed, a volume of at least 100uL buffer is always present on top of the SLBs all along the process. Successively, Streptavidin, biotinylated Phalloidin, actin (pre-polymerized aside) and labelled Phalloidin are added to the initial 100uL buffer along SLB functionalization. It increases the total volume of Buffer over the SLB to 150uL. This is now described more in details in the new version of the Material and Methods section.

- figures have to be checked carefully. Fig 4C/D should show the region of the zoom. Fig 5 G is missing units at the axis label, Fig 5 E/F should use another colour code than Red/Green as this is not suitable for colour blind people.

We thank the reviewer for pointing this out. The region of the zoom has been shown in the new Figure 4C & D. The figure 5 has been changed due to the integration of new data, and we have changed the color code of the figure for graphs, images and schemes. We paid particular attention in avoiding the red/green code. Finally, we also tried to homogenize all the figures in terms of color codes, statistics representation etc... We hope this makes the figures easier to read now.

-the writing needs in some regions improvement. In the beginning of results, the authors should name the Rac1 and CDC42 inhibitors used, when writing of IC50 values, they should say 150 μ M of what. Page 5, 'Quantification of HIV-1 p24 release showed a 50% significant increase ...' doesn't make much sense. Page 6, doubling time gets faster or shorter, not lower. And many more such instances. - Page 7: '... f-actin mean intensity of gag+ area of value 0.9435...' of what? Given the std of 0.16, there are too many digits given. The precision doesn't allow to write all four decimals. It should be 0.94 +/- 0.16 and then a unit should be written.

Thanks to the reviewer for these comments: we have made all changes accordingly in the main revised manuscript. We kept the results to be in between 2 and 3 (max) significant digits, which are more connected to our measurement precisions.

Reviewer #3 (Remarks to the Author):

Dibsy and coworkers describe studies that represent a logical and appropriate follow-up to their 2015 J. Virology paper (PMID: 26018170). In the previous study, the group reported that activated Rac1 and the IRSp53-Wave2-Arp2/3 signaling pathway were involved in HIV-1 Gag membrane localization and particle release in T-cells as well as a Gag-dependent role for actin debranching and polymerization. In this current report, the authors have assessed if Arp2/3 mediated branched actin is involved in HIV assembly at the membrane of infected T-cells. They show that actin debranching increases HIV particle release as well as the number of HIV assembly clusters at the membrane. They further show that Gag assembly is found in areas with less F-actin, which they further demonstrated with in vitro studies. Arpin (a Arp2/3/ inhibitor) was recruited by Gag and promoted virus assembly.

Major comments:

1. The general topic understudy in this manuscript is both timely and important. Here, the authors provide a logical follow up to their previous work and have uncovered additional details to help better explain how the IRSp53-Wave2-Arp2/3 signaling pathway is involved in HIV assembly. The observation of virus release being associated with actin debranching is important. As outlined below, there are number perceived weaknesses that limit the interpretative power of the dataset.

We thank the reviewer for his/her comment. We have now tried to improve the manuscript in the view of his/her comment to make it stronger.

2. The use of drugs that affect actin dynamics (ie, Fig 2 and Jaslakinolide, LatrunculinB), while are being investigated at concentrations that appear to minimize cell toxicity), would be enhanced if they were complemented with other experimental approaches as the drugs could have off-target effects that leading to the observations being made.

We agree with the reviewer's concerns that these drugs (Lat B and Jasplakinolide) could have off-target effects (Figure 1). However, these are well-established commercial drugs found in the literature in order to manipulate the polymerization/depolymerization equilibrium of F-actin (as

referred in the manuscript, and verify the drug effect on F-actin using imaging (new Figure 1B, Figure 2A) or cytometry (Supplementary Figure 3A). It is well known that Jasplakinolide, for example, binds specifically and strongly ($K_d \sim 10$ nM) to actin filaments, leading to less probable off-target effects of the drug. In the case of the CK666 drug (Figure 2), we controlled that our effect was directly due to the Arp2/3 debranching using an analog CK689, known to be the negative control. Although off-target effect can still occur, using a control drug decrease their probability to account for the specific effect observed. In addition, as another experimental approach, we show that Arpin, a cellular protein inhibiting the Arp2/3 complex lead to the same effect (Figure 6), thus, we can reasonably rule out off-target effects.

3. The latrunculin data seems somewhat confusing. If decreasing F-actin lowers particle production, it would seem to go against the idea that increasing actin debranching helps assembly. They seem to have a more complex model of the role of "actin treadmilling". This data and the applicability of the complex treadmilling modeling needs to be better clarified.

We thank the reviewer for pointing this out. We agree that the word "actin treadmilling" was confusing, so we removed this word from the revised manuscript and investigate more on the effect of Latrunculin B on HIV-1 particle release in T cells (new Figure 1). In order to respond to reviewer's comments, new experiments at different concentrations of LatB on viral particle release were performed and added in new Figure 1 (more replicates for alphaLisa, immunoblots and F-actin imaging). Interestingly, we found an effect of LatB on HIV-1(i)GFP release from Jurkat T cells infected single round which is enhanced upon LatB treatment. Our new results show that upon F-actin depolymerization by LatB, thus creating less dense cortical actin (see Figure 1B), virus release increases (as shown by western blot on purified virions – Figure 1C). As LatB and CK666 are 2 different drugs targeting F-actin which end results is to decrease cell F-actin content, we are confident to say that less F-actin favors HIV-1 release (new figure 1 and figure 2)

In contrast, there is a drastic inhibitory effect on HIV release upon Jasplakinolide treatment (Figure 1, Figure Supplementary 2B).

So we have clarify the Figure 1 and the text accordingly (pages 4-5) in that sense to decrease any confusion to the readers.

4. The experiments with CK666 (Fig 3 & 4), while informative, raise a similar concern that off-target effects from the drugs could be influencing the results being reported.

We thank the reviewer for his/her remark similar to the point #2 he/she previously raised.

By using CK689 analog, known also to bind to Arp2/3 complex without affecting it, we reasonably eliminate potential off-target effects of CK666 on HIV-1 particle release increase. In addition, CK666 and Arpin, both targeting and inhibiting the Arp2/3 complex have similar effects on HIV-1 particle release (as shown by the results of Figure 2 and 6).

5. Fig 5 uses an in vitro model to investigate Gag clustering on model membranes. The strength of this model is that they can manipulate conditions to access the effects on labeled Gag clustering. It is not clear whether actin formation on the model membranes is relevant to what is seen along the plasma membrane of cells. The general data shown tends to correlate with actin influencing Gag clustering, but it is unclear how well this in vitro model simulates sites of Gag clustering in cells regarding form of actin and membrane lipid composition.

We agree with this comment, which is somehow similar to the comment #3 of Rev#2. It is true that our proposed diffusion restriction model to explain decrease in HIV-1 assembly in dense branching actin area cells is mainly supported by the *in vitro* experiments, and we have no direct proof that the actin organization of our reconstructed supported lipid bilayers/actin model perfectly mimics the overall organization of the plasma membrane and its underneath cortical actin. As mentioned this is the limits of *in vitro* models. However, to circumvent that issue, we have now performed PI(4,5)P2 mobility experiments using the same spot variation FCS technique as we used with SLBs, and we have measured the PI(4,5)P2 diffusion coefficient in Jurkat T-cells exposed or not to CK-666 drug treatment and labelled with the lipid analogue Atto647N-PI(4,5)P2 (the one used also to monitor diffusion in SLBs).

The new results are shown in Fig.5C and D of the manuscript. Interestingly, we found that PI(4,5)P2 diffusion coefficient significantly increase (almost 2 times) in cells treated with CK666 compared to the DMSO only treated cells. This change is perfectly in line to the one we previously observed between SLB containing either no actin on top of the membrane or 4.4 μM of actin.

In addition, it has been previously shown that, when plotting the so called “FCS diffusion laws”, negative values intercept at null waist was a signature of fences and meshes (as in branched actin cytoskeleton) restriction of diffusion (Warweczinieck et al., BJ 2005, ref 43 of our revised manuscript). Interestingly this is exactly what we observe here in DMSO exposed cells and SLB covered with increasing actin concentration. This strongly suggest that, at least for PI(4,5)P2 lateral mobility, our SLBs are nicely mimicking cells plasma membrane and underneath actin cytoskeleton organization, although certainly not perfectly. Finally, thanks again to the FCS diffusion law, we can also estimate the size of the meshwork which is impeding the diffusion by measuring the value of w^2 at $\tau=0$ (x-axis intercept of the diffusion law). In the case of the 4.4 μM actin with 1% mol biotin SLBs, we found here $\langle w^2(t=0) \rangle = 0.017 \mu\text{m}^2$ (average of 3 different SLBs) (Supplementary Figure 8B), which correspond (for a square mesh) to an average value of 130 nm side length for a mesh unit. When performing the same analysis in Jurkat T cells, we found 180 nm (Supplementary Figure 8A). Although different, these two meshes size are similar, suggesting again that, the effect on molecular mobility occurring in cells is nicely mimicked by the SLB/actin *in vitro* system.

We want to acknowledge here rev#3 as well as rev#2 for their comment on this point, since both clearly helped us to reinforce our proposed mechanism.

6. The authors targeted the expression of Arpin (which is a actin debranching factor) in Fig 6. Reducing Arpin expression by 40% resulted in a very modest reduction in HIV release. The authors state that this results in a reduction of 30% in HIV release, while the actual histogram in Fig 6C looks no better than 20%, and with the variability of the assay this difference may be less than this. Overall, the trend supports the conclusion that the authors are seeking to draw. However, the data is not strongly supportive of the conclusion.

We thank the reviewer for that comment that give us a chance to clarify. For a 40% decrease in Arpin expression (Fig.6B), the resulting effect on HIV release was a decrease of $25 \pm 6\%$ (with a significant p-value=0.0462), which indeed is striking if you consider the ratio gene inhibition over particle release inhibition (which is the relative difference) (now presented on the new Fig. 6D). This was highly reproducible (please see table of the raw data analysis below). Unfortunately, we could not emphasized better this phenotype (using siRNA without cell toxicity). So, in this condition, the siRNA treated infected T cells were alive and tolerating Arpin gene knock-down to monitor significant virus release.

This pointed out, as suggested in Figure 4A/B, that Gag assembly clusters are located in plasma membrane area enriched but also non-enriched in F-actin. So, our hypothesis is that when only Gag is located in F-actin(+) area, it would require Arpin to inhibit branched actin in order for Gag release to proceed further. This would explain the partial dependency of Gag for Arpin.

So, we affirm that our data (Figure 6) are supportive to the conclusion.

Raw data release	siCTRL	siArpin	Normalized data release	siCTRL	siArpin	
	60	40		100	63	
	58	50		100	79	
	71	52		100	83	
Mean	63	47		100	75	25% decrease

7. It is unclear if the Gag clustering observed by the authors can be differentiated from released particles immobilized under the cell in their microscopy and in their TIFR images. It would be expected that released particles would swamp out any population of Gag clustering at the membrane. The rate at which Gag clustering occurs is quite fast, and the odds of detecting Gag clustering prior to particle release would be challenging.

We agree that suggesting a clear distinction of clusters representing on going assemblies and clusters representing fully assembled particles released or not from the cell will be an over-interpretation of the results, even more in the case of simple TIRF imaging. However, if the released particles were swamping out any population of Gag clustering, we would expect that the size distribution of the identified clusters, thanks to our SMLM imaging, will exhibit the same size distribution than that of the virus or viral particles (Gag-VLP) released from the cells.

In Floderer et al. (figure 2 of ref 45 of our main manuscript), in Jurkat T-cells, we showed that it is systematically not the case (see graph C of the figure 2 from Floderer et al. (45)), for WT Gag-VLP released from the cells. We found the mean size of the release WT Gag-VLP to be 139 ± 7 nm in diameter. Here, although using Gag(i)GFP STORM microscopy, we found the mean size of clusters at the cell membrane of Jurkat T cells to be respectively 94 nm and 107nm for DMSO and CK666 treated cells (Figure 3), which is clearly below the 140 nm previously measured for Gag-VLP release from Jurkat T cells. This is now illustrated in Fig.3C and clarify in the result section (page 9) of the revised manuscript.

Additionally, in the protocol we used here, the cells are seeded only 30 min before fixation. This 30 min correspond to the time course of a full cycle of viral particle release (from 20 to 30 min, depending on the authors, Floderer et al., (45), Jouvenet et al., Nature 2008 (doi.org/10.1038/nature06998), PNAS 2009 (doi.org/10.1073/pnas.0907364106) Ivachenko et al., Plos Path 2009 (doi.org/10.1371/journal.ppat.1000652)). Therefore at the given time of fixation, we shall have all the state of assembly with equivalent probability. Knowing that the time course of assembly is found to represent 20 to 50% of the total time needed to make and release a full virus, one can reasonably expect that the clusters we identified are at worst representing 13% and more probably 30% (for 20 min assembly) to 50% (for 30 min assembly) of the total clusters observed.

Minor comments

1. The fixation protocol for the microscopy imaging is a bit concerning. 4% PFA can result in a significant loss of soluble proteins, like Gag and G-actin.

We tried several fixation protocols for performing high resolution microscopy (indeed 4%PFA alone is not a good fixative) that's why we use a combination of 4%PFA+ 4% sucrose which preserved very well F-actin and membrane structures for super-resolution imaging. In addition, we imaged Gag at the cell membrane and F-actin but not G-actin, using A647-Phalloidin, and try to minimize soluble Gag imaging which is considered as background. So, our experimental condition that we set up for imaging are very good for imaging Gag clusters at the cell membrane and cortical actin.

2. Cytoplasmic HIV Gag diffuses slowly in part due to the presence of the cytoskeleton. Disrupting the cytoskeleton might lead to a faster loss of soluble Gag from cells, then normalizing intensity as described by the others in several instances could lead to the observation of relatively more Gag at the membrane. The diffusion timescale for Gag to be released from a damaged cell is probably several minutes or more, which is about the time used in fixation.

We thank the reviewer for this comment, which enlighten a possible confusing sentence in the manuscript. Indeed, the fluorescence intensities in figure 3A or figure 4B are not normalized to the Gag intensity. They are normalized to the total actin intensity. We have tried to express this more clearly in the text to avoid any confusion for the reader.

However, in order to control whether Gag diffusion was strongly impacted by actin network, we performed FCS at single waist measurements of our GFP-Gag in cells treated or not with CK-666. We took cells where the concentration of Gag is sufficient to allow for assembly and particle release (cite paper of J. Mueller). As shown in the figure below we did not observe strong difference between CK-666 treated or not diffusion coefficient of Gag in the cytosol. We found $\langle D_{CK666} \rangle = 4.0 \pm 0.1$ (mean \pm sem) and $\langle D_{DMSO} \rangle = 5.8 \pm 0.1$ (mean \pm sem). Although this could be surprising, it is worth to note that D_{DMSO} value is found to be similar to the one observed in Hendrix et al., doi.org/10.1083/jcb.201504006, using RICS technique to measure diffusion, this reinforces the viability of our measurements. Moreover, partial disruption of cytoskeleton, as it is induced by our CK-666 drug treatment might not be sufficient to strongly affect the diffusion as the effect is clearly not linear with the size of the diffusing object (see Dauty et al. [10.1074/jbc.M412374200](https://doi.org/10.1074/jbc.M412374200) for example)

T cells are approximately 10 µm diameter, where the volume is mostly occupied by the nucleus (6-8 µm diameter), suggesting that such a cytosolic diffusion coefficient is not a limiting factor to reach the membrane, as it will be achieved within seconds in both cases (CK666 treated or not). Finally, during fixation and cell permeation, the flux of molecules crossing the membranes can be written as $\Phi = -P(C_{out} - C_{cyt})$ (Similar to Fick's first law) with $P(t) = 2 A_p d_{TM} \sqrt{D/\pi t}$ (deduced from Langmuir-schaeffer law of adsorption, see [doi: 10.1021/ja01290a091](https://doi.org/10.1021/ja01290a091)), where A_p is the size of the membrane pore (this is supposed to be similar in both cases as it mainly depends on the permeabilization process), d_{TM} is the diffusion across the membrane, which is supposed to be similar in both cases as we observe the same molecular specie. Therefore, the change in concentration outside of the cell will scale with the square root of the cytosolic diffusion coefficient, which here is

not only almost identical in terms of value, but moreover lower for CK666 treated cells than for not treated cells. For all these reasons, we believe that the changes in intensity we observe in Gag, when they are present are not due to different leaking fluxes for CK666 vs DMSO treated cells during fixation.

3. More details are needed for the model for how diffusion of HIV Gag on membranes can influence the initiation of Gag clustering. The dynamics of this process would appear to be fairly complex, and it seems to be appropriate to incorporate a reaction diffusion model.

We thank the reviewer for this comment, it is clear that initiation of the self-assembly is a reaction-diffusion process. It is not our aim in this paper to model this kinetics, since it has been published in different papers, most notably in Zang et al. *Phys Rev E* (10.1103/PhysRevE.78.051903). In this paper, the initiation of the assembly is linked to collision-based aggregation and diffusion, i.e. the reaction diffusion model. Based on the diffusion-limited aggregation model (*D. F. Evans and H. Wennerström, The Colloidal Domain: Where Physics, Chemistry, Biology, and Technology Meet, 2nd ed. Wiley-VCH, New York, 1999.*) it is shown that the assembly initiation is highly sensitive to monomer diffusion coefficient at the membrane. The second part of the assembly process is similar to what is observed in biological condensate and the growing of the assembly belongs to the Lifshitz-Slezov LS regime. Addition of monomer depends on the available pool which itself has a characteristic time which only depends on the binding energy of monomers. Based on this model, it is clear that actin will play a minor role on the second part of the process unless the self-binding energy of Gag is modulated by the presence of actin. We don't see any simple process that could account for this.

Additionally, in the first part of the assembly, the reaction-diffusion process should not be strongly affected by actin by a different process than diffusion itself, unless again one expects that Gag will systematically bind to actin thanks to its Gag-Gag binding domain, competing for the reaction process, otherwise, again, this is diffusion that will mainly be concerned, as suggested in Zang et al.

We have now introduced the Zhang et al. reference (ref 40) discussing the impact of diffusion on the initiation of assembly and we have added a sentence at the beginning of the corresponding results section.

REVIEWERS' COMMENTS

Reviewer #1 (Remarks to the Author):

The authors have addressed many of the reviewer comments. A few additional issues are listed below. More importantly, concerns remain about the very small magnitude of many of the reported effects.

1. Fig. S1D. Why is virion GagGFP larger (more slowly migrating) than cell-associated GagGFP?
2. Fig. S1E. It is not possible to figure out what was done here.
3. Fig. S1F. It is not clear what this panel is showing.
4. Line 318: what does "laser waist parking" mean?
5. Throughout, the MS requires major English editing.

Reviewer #2 (Remarks to the Author):

The revised manuscript 'HIV-1 diverts actin debranching mechanisms for particle assembly and release in CD4 T lymphocytes' by Dibsby et al. is well improved in its clarity and strength of experimental data compared to the previous version. The authors have made a commendable effort addressing all reviewer questions and concerns with additional experimental data and clear explanations of their reasoning. Reading the very well prepared rebuttal was a great pleasure and was very informative.

For example, providing now the reasoning for performing actin polymerization on the SLBs at 4C makes this step now very understandable and will be helpful for other labs. The detailed measures of protein mobility using spot size variation FCS are formidable and convincing.

However, some conclusions should be drawn a bit more carefully given the presented data.

From the actin perturbation experiments using drugs should be formulated more carefully given the presented data. In Lymphocytes, LatB leads to an increase and Jasp to a (slight) decrease of viral particle release, which is argued to be linked to actin debranching and its stabilization, respectively. Since both drugs affect actin turnover in general and since the authors don't show that branched actin is specifically affected, this claim is not well founded.

Further, the results on primary lymphocytes seem to partially contradict the authors conclusion, because here now Lat B leads to a reduction of viral particle release. The authors should discuss this difference and not simply conclude the paragraph in the results section with the statement that actin debranching leads to increase viral release.

Regarding the in vitro experiments, it would be good to emphasize that the minimal model does not imply that it directly reflects the situation in cells. It simply shows that a change in actin network density can indeed lead to changes in Gag clustering due to an altered Gag diffusion along the membrane.

Taken together, the manuscript is suitable for publication with minor corrections.

REVIEWERS' COMMENTS

Please find enclosed our responses to the reviewers' comments (in blue).

Reviewer #1 (Remarks to the Author):

The authors have addressed many of the reviewer comments. A few additional issues are listed below. More importantly, concerns remain about the very small magnitude of many of the reported effects.

1. Fig. S1D. Why is virion GagGFP larger (more slowly migrating) than cell-associated GagGFP?

Reviewer 1 is correct that virion GagGFP proteins appeared to migrate slower. It is currently unclear to us how the internal environment within a virion may differ from the cell lysate of the Gag producing cells and affect Gag migration in gels/western blots. One hypothesis could be some post-translational modification of Gag inside the virions (like phosphorylation, sumoylation or ubiquitinylation) as it has been reported before.

2. Fig. S1E. It is not possible to figure out what was done here.

To clarify, we have now provided additional explanation in the correspondent legend:

E) Histograms showing HIV-1 titration. Viruses were produced from 293T HEK cell line transfected with pNL4.3 WT or VSV-G pseudotyped pNL4.3(i)GFPdeltaEnv.

We apologize with the confusions. Please see Supplemental Information, Figure S1. Purified viruses were used to infect Jurkat T lymphocytes (see method). Quantification of purified HIV-1 using an alphaLISA anti-CAP24 (ng/ml). Three independent HIV-1 productions are shown: #1, #2 and #3. Each HIV-1 production was compared to the titer of the WT virus stock.

3. Fig. S1F. It is not clear what this panel is showing.

To clarify, we add more explanation in the correspondent legend:

F) Flow cytometry showing the level of infectivity of VSV-G pseudotyped HIV-1(i)GFPdeltaEnv virus on CD4+ T Jurkat cells (n=20 000 events).

Infectivity is quantified by measuring the intensity of GFP in non-infected cells in comparison with HIV-1(i)GFP infected cells. The panel shows the percentage of GFP(+) cells, represented in the cell population with high GFP intensity as compared to the control. The percentage of GFP(+) cells represents the % of infected cells.

4. Line 318: what does "laser waist parking" mean?

The sentence "Indeed, we have recently shown that small deviations of the laser waist parking from the cell membrane surface lead to rapid increase in apparent membrane bound diffusion coefficient⁴² that might lead to mis-interpretation" has been rewritten to :

“Indeed, we have recently shown that small deviations of the axial laser position from the cell membrane surface generate increased values of the apparent membrane bound diffusion coefficient that might lead to mis-interpretation.” (lanes 317-319, page 11).

5. Throughout, the MS requires major English editing.

The manuscript has now been read and edited by a native English speaker.

Reviewer #2 (Remarks to the Author):

The revised manuscript 'HIV-1 diverts actin debranching mechanisms for particle assembly and release in CD4 T lymphocytes' by Dibsby et al. is well improved in its clarity and strength of experimental data compared to the previous version. The authors have made a commendable effort addressing all reviewer questions and concerns with additional experimental data and clear explanations of their reasoning. Reading the very well prepared rebuttal was a great pleasure and was very informative.

We thank the reviewer for this comment.

For example, providing now the reasoning for performing actin polymerization on the SLBs at 4C makes this step now very understandable and will be helpful for other labs. The detailed measures of protein mobility using spot size variation FCS are formidable and convincing.

We deeply thank the reviewer for this comment.

However, some conclusions should be drawn a bit more carefully given the presented data. From the actin perturbation experiments using drugs should be formulated more carefully given the presented data. In Lymphocytes, LatB leads to an increase and Jasp to a (slight) decrease of viral particle release, which is argued to be linked to actin debranching and its stabilization, respectively. Since both drugs affect actin turnover in general and since the authors don't show that branched actin is specifically affected, this claim is not well founded.

The title is proposed to be changed based on this comment. We agree with the proposed new title:

« HIV-1 diverts cortical actin for particle assembly and release. »

Further, the results on primary lymphocytes seem to partially contradict the authors conclusion, because here now Lat B leads to a reduction of viral particle release. The authors should discuss this difference and not simply conclude the paragraph in the results section with the statement that actin debranching leads to increase viral release.

We thank the reviewer for this comment.

The sentence “However, the effect of F-actin depolymerization, using Latrunculin B treatment, on HIV-1 release from infected PBLs can vary depending on drug concentrations when using p24 alphaLISA immunoassay technique. “was already written in the result section (lane 154-156, page 6).

Now another sentence has been added in the Discussion section, lane 444-447:

Indeed, an opposite effect of Latrunculin B on HIV-1 production in primary lymphocytes has been reported in the literature compared to Jurkat T cells. Similar to our data, published results showed a

decrease in HIV-1 production from primary lymphocytes treated with Latrunculin B and an increase in the case of Jurkat T cells (see reference [56]). This apparent contradiction is not currently explained.

Ref [56]: Cell-to-Cell Transmission Can Overcome Multiple Donor and Target Cell Barriers Imposed on Cell-Free HIV. Peng Zhong ,Luis M. Agosto ,Anna Ilinskaya,Batsukh Dorjbal,Rosaline Truong,David Derse,Pradeep D. Uchil,Gisela Heidecker,Walther Mothes. PLoS One. 2013;8(1):e53138. doi: 10.1371/journal.pone.0053138. Epub 2013 Jan 7.

Regarding the in vitro experiments, it would be good to emphasize that the minimal model does not imply that it directly reflects the situation in cells. It simply shows that a change in actin network density can indeed lead to changes in Gag clustering due to an altered Gag diffusion along the membrane.

To take into consideration the suggestion of the reviewer, the sentence “This result is remarkably in line with what we observe in infected cells (Fig.4).” has been changed to “Although the use of minimal in vitro system does not imply that it directly reflects the situation in cells, this result is remarkably in line with what we observe in infected cells (Fig.4).” (lanes 477-478, page 16).

Taken together, the manuscript is suitable for publication with minor corrections.